A new view on the morphology and phylogeny of eugregarines suggested by the evidence from the gregarine Ancora sagittata (Leuckart, 1860) Labbé, 1899 (Apicomplexa: Eugregarinida)

http://orcid.org/0000-0003-2478-9301 Simdyanov Timur G. 1 tgsimd@gmail.com
Guillou Laure 2 3
Diakin Andrei Y. 4
Mikhailov Kirill V. 5 6
Schrével Joseph 7 8
http://orcid.org/0000-0002-3299-9950 Aleoshin Vladimir V. 5 6
1 Faculty of Biology, Department of Invertebrate Zoology, Lomonosov Moscow State University , Moscow , Russian Federation
2 UMR 7144, Laboratoire Adaptation et Diversité en Milieu Marin, CNRS , Paris, Roscoff , France
3 UMR 7144, Station Biologique de Roscoff, CNRS, Sorbonne Universités, Université Pierre et Marie Curie - Paris 6 , Paris, Roscoff , France
4 Faculty of Science, Department of Botany and Zoology, Masaryk University , Brno , Czech Republic
5 Belozersky Institute of Physico-Chemical Biology, Lomonosov Moscow State University , Moscow , Russian Federation
6 Institute for Information Transmission Problems, Russian Academy of Sciences , Moscow , Russian Federation
7 CNRS 7245, Molécules de Communication et Adaptation of Micro-organisms , Paris , France
8 Muséum National d’Histoire Naturelle, UMR 7245, Sorbonne Universités , Paris , France
Reimer James
Electronic publication date: 2017 May 30
Publication date: 2017
Volume: 5
Electronic Location ID: e3354
Received 2016 Aug 25; Accepted 2017 Apr 26
Copyright: © 2017 Simdyanov et al.
Copyright year: 2017
Copyright holder: Simdyanov et al.
License: This is an open access article distributed under the terms of the Creative Commons Attribution License, which permits unrestricted use, distribution, reproduction and adaptation in any medium and for any purpose provided that it is properly attributed. For attribution, the original author(s), title, publication source (PeerJ) and either DOI or URL of the article must be cited.
License URL: https://creativecommons.org/licenses/by/4.0/

Keywords: Apicomplexa, Marine gregarines, Ultrastructure, SSU and LSU rDNA, Environmental DNA sequences, Phylogeny, Taxonomy

Funding: Russian Foundation for Basic Research project No. 15-29-02601 ECO-NET project 2131QM (Égide, France) The European Project MaCuMBa FP7-KBBE-2012-6-311975 The French governmental ANR Agency ANR-10-LABX-0003 BCDiv, ANR-11-IDEX-0004-02, and ANR HAPAR 2014 défi 1 The Interdisciplinary Program of the MNHN (ATM-Emergence des clades, des biotes et des cultures) Czech Science Foundation project No. GBP505/12/G112 (ECIP) Russian Science Foundation 14-50-00029 This work was supported by the Russian Foundation for Basic Research (project No. 15-29-02601), ECO-NET project 2131QM (Égide, France), the European Project MaCuMBa (FP7-KBBE-2012-6-311975), the French governmental ANR Agency under ANR-10-LABX-0003 BCDiv, ANR-11-IDEX-0004-02, and ANR HAPAR 2014 défi 1, the Interdisciplinary Program of the MNHN (ATM-Emergence des clades, des biotes et des cultures), and the Czech Science Foundation, project No. GBP505/12/G112 (ECIP). The phylogenetic analyses of SSU rDNA presented in this study were supported by the Russian Science Foundation, project No. 14-50-00029. There was no additional external funding received for this study. The funders had no role in study design, data collection and analysis, decision to publish, or preparation of the manuscript.

==============================
Background

Gregarines are a group of early branching Apicomplexa parasitizing invertebrate animals. Despite their wide distribution and relevance to the understanding the phylogenesis of apicomplexans, gregarines remain understudied: light microscopy data are insufficient for classification, and electron microscopy and molecular data are fragmentary and overlap only partially.

Methods

Scanning and transmission electron microscopy, PCR, DNA cloning and sequencing (Sanger and NGS), molecular phylogenetic analyses using ribosomal RNA genes (18S (SSU), 5.8S, and 28S (LSU) ribosomal DNAs (rDNAs)).

Results and Discussion

We present the results of an ultrastructural and molecular phylogenetic study on the marine gregarine Ancora sagittata from the polychaete Capitella capitata followed by evolutionary and taxonomic synthesis of the morphological and molecular phylogenetic evidence on eugregarines. The ultrastructure of Ancora sagittata generally corresponds to that of other eugregarines, but reveals some differences in epicytic folds (crests) and attachment apparatus to gregarines in the family Lecudinidae, where Ancora sagittata has been classified. Molecular phylogenetic trees based on SSU (18S) rDNA reveal several robust clades (superfamilies) of eugregarines, including Ancoroidea superfam. nov., which comprises two families (Ancoridae fam. nov. and Polyplicariidae) and branches separately from the Lecudinidae; thus, all representatives of Ancoroidea are here officially removed from the Lecudinidae. Analysis of sequence data also points to possible cryptic species within Ancora sagittata and the inclusion of numerous environmental sequences from anoxic habitats within the Ancoroidea. LSU (28S) rDNA phylogenies, unlike the analysis of SSU rDNA alone, recover a well-supported monophyly of the gregarines involved (eugregarines), although this conclusion is currently limited by sparse taxon sampling and the presence of fast-evolving sequences in some species. Comparative morphological analyses of gregarine teguments and attachment organelles lead us to revise their terminology. The terms “longitudinal folds” and “mucron” are restricted to archigregarines, whereas the terms “epicystic crests” and “epimerite” are proposed to describe the candidate synapomorphies of eugregarines, which, consequently, are considered as a monophyletic group. Abolishing the suborders Aseptata and Septata, incorporating neogregarines into the Eugregarinida, and treating the major molecular phylogenetic lineages of eugregarines as superfamilies appear as the best way of reconciling recent morphological and molecular evidence. Accordingly, the diagnosis of the order Eugregarinida Léger, 1900 is updated.

Introduction

The Apicomplexa is a group of unicellular eukaryotes within the Alveolata encompassing parasites of humans and animals. Some apicomplexans are well studied (e.g., human pathogens such as Plasmodium, Toxoplasma, and Cryptosporidium), while early branching representatives, such as gregarines, are far less well known. Gregarines are obligate parasites of invertebrate animals: various groups of worms, molluscs, arthropods (aquatic and terrestrial), echinoderms, and tunicates. The large majority of gregarines are monoxenous (have a single invertebrate host) and parasitize in the gut of their hosts, where they are commonly found as epicellular feeding stages, the trophozoites, which are conspicuous due to their large size (usually from 200 to 600 μm). Because of their minor economic importance, gregarines are poorly studied despite their widespread distribution and relevance to the reconstruction of the evolutionary history of apicomplexans.

The taxonomy and phylogeny of the gregarines remains largely incomplete (Grassé, 1953; Levine, 1985, 1988; Perkins et al., 2000) due to uneven scrutiny: light microscopic data cannot sustain a reliable classification, electron microscopy and molecular phylogenetic data are fragmentary, and, additionally, sets of features that have been examined using different methods overlap only partially (see Discussion). Gregarine orders differ by their life cycles, which include sexual (gamogony) and asexual (merogony and sporogony) reproductions. Sexual reproduction in gregarines (Grassé, 1953; Schrével et al., 2013) is initiated by syzygy (the association of gamonts, usually two of them) and followed by the production of a surrounding gametocyst, which is typical only for gregarines and likely represents a synapomorphy for the group (Frolov, 1991). The large majority of gregarines, which are classified within the order Eugregarinida Léger, 1900, have lost merogony, while some others (former order Schizogregarinida Léger, 1900) retain it.

The most productive taxonomical scheme of the gregarines is based on Grassé’s hypothesis about their co-evolution with their hosts (Grassé, 1953). Grassé divided Schizogregarinida into two orders: Archigregarinida Grassé, 1953 and Neogregarinida Grassé, 1953. Archigregarines parasitize marine invertebrates, mainly polychaetes and sipunculids. Neogregarines parasitize insects (intestine, Malpighian tubules, and fat body) and Grassé suggested that they are derived from various representatives of the eugregarine family Actinocephalidae (parasites of insects), by the secondary gain of merogony. The third order, the already mentioned Eugregarinida Léger, 1900, is the most diverse group of gregarines infecting a broad range of invertebrate hosts.

The current gregarine classification (Levine, 1988; Perkins et al., 2000) relies chiefly on the light-microscopy of trophozoites and life cycle features (absence or presence of merogony), discarding Grassé’s co-evolutionary approach. It also ignores results of scanning electron microscopy (SEM) and transmission electron microscopy (TEM) studies, which have revealed distinct differences between Grassé’s gregarine orders in the structure of the cortex and attachment apparatus, especially between archi- and eugregarines (Schrével, 1968; Vivier, 1968; Vávra & Small, 1969; Vivier et al., 1970; Schrével, 1971a, 1971b; Simdyanov & Kuvardina, 2007; Schrével et al., 1983; see Discussion). As a result, a portion of the archigregarines were reassigned to eugregarines (Levine, 1985, 1988), which in turn were divided into two main suborders: Septata Lankester, 1885 and Aseptata Chakravarty, 1960. Aseptate eugregarines (e.g., the families Lecudinidae and Urosporidae) chiefly infect marine invertebrates and considered plesiomorphic representatives of the order (Grassé, 1953; Perkins et al., 2000; Schrével & Desportes, 2013b). Septate gregarines are widespread parasites of aquatic and terrestrial arthropods and considered evolutionarily derived: they possess one or more light-refracting septum, which separates the trophozoite into two compartments: a smaller protomerite and larger deutomerite, where the nucleus is located.

Molecular phylogenetic studies of gregarines are limited in sampling and largely rely on small subunit (SSU or 18S) ribosomal DNA (rDNA) sequences (Carreno, Martin & Barta, 1999; Leander, Clopton & Keeling, 2003; Leander, Harper & Keeling, 2003; Leander et al., 2006; Leander, 2007; Lepelletier et al., 2014; Rueckert & Leander, 2008, 2009, 2010; Clopton, 2009; Rueckert, Chantangsi & Leander, 2010; Rueckert et al., 2011, 2015; Rueckert, Villette & Leander, 2011; Rueckert, Wakeman & Leander, 2013; Wakeman & Leander, 2013a; Wakeman & Leander, 2012, 2013b; Wakeman, Heintzelman & Leander, 2014; Wakeman et al., 2014; Diakin, Wakeman & Valigurová, 2017). Gregarines have been also detected in environmental sequence surveys from various marine and freshwater samples, possibly because oocysts are stable in the environment (Rueckert et al., 2011; Janouškovec et al., 2015). A large majority of these environmental sequences cannot be taxonomically assigned to a specific gregarine family. Because many gregarine SSU rDNA sequences are fast evolving and form long branches in molecular phylogenies, the entire group and its orders are not recognized as monophyletic. This has led to the proposal that eugregarines are polyphyletic (Cavalier-Smith, 2014) and their shared key ultrastructural characteristics have been acquired convergently (see Discussion).

In this work, we characterize the aseptate eugregarine Ancora sagittata (Leuckart, 1861) Labbé, 1899, an intestinal parasite of the marine polychaete worm Capitella capitata Fabricius, 1780, a widely distributed and abundant inhabitant of oxygen-depleted substrates. Ancora sagittata has been classified as a member of Lecudinidae Kamm, 1922, the largest family of marine aseptate eugregarines (containing ∼30 genera and >160 named species). The taxonomy of Lecudinidae is nevertheless controversial and the family may not represent a natural group (Levine, 1977, 1985, 1988; Perkins et al., 2000).

Trophozoites of Ancora sagittata have a characteristic anchor-like appearance (Labbé, 1899; Perkins et al., 2000) and their structure, growth, and development were previously observed by light microscopy (Cecconi, 1905; Hasselmann, 1927). The sexual reproduction of Ancora sagittata is little understood (Hasselmann, 1927) and syzygy in this species has never been observed. Neither ultrastructural nor sequence data are currently available for the parasite. Here, we undertook an integrated study of the Ancora sagittata morphology, ultrastructure, and molecular phylogeny by using rDNA: SSU (18S), 5.8S, and LSU (28S). We revealed that Ancora sagittata represents a deep molecular phylogenetic lineage of eugregarines independent of the Lecudinidae in spite of their morphological similarities. This finding led us to re-evaluate and reconcile ultrastructural and molecular evidence for eugregarines and, relying on this combined approach, amend conventional views on eugregarine phylogeny and taxonomy.

Materials and Methods

Collection, isolation, and light microscopy

Trophozoites of Ancora sagittata (Leuckart, 1860) Labbé, 1899 were isolated from the intestine of the polychaete worms Capitella capitata Fabricius, 1780 collected in 2006–2011 from two sites: (i) littoral of the beach of L’Aber, the coastal zone of the English Channel near Station Biologique de Roscoff, Roscoff, France (48°42′45″N, 4°00′05″W) and (ii) a sublittoral habitat at White Sea Biological Station (WSBS) of Lomonosov Moscow State University, Velikaya Salma Straight, Kandalaksha Gulf of White Sea, Russia (66°33′12″N, 33°06′17″E).

The gregarines were isolated by breaking the host body and intestine with fine tip needles under a stereomicroscope (Olympus SZ40, Olympus, Tokyo, Japan, or MBS-1, LOMO, St. Petersburg, Russia). The released parasites or small fragments of host gut with attached gregarines were rinsed with filtered seawater by using thin glass pipettes and then photographed under Leica DM 2000, Leica DM 2500, or Leica DM5000 light microscopes with Leica DFC 420 cameras (Leica Microsystems, Wetzlar, Germany), or fixed for electron microscopy, or subjected to DNA extraction.

Electron microscopy

The structure of the gregarines Ancora sagittata from WSBS was studied by SEM and TEM. For both methods, the individual gregarines or small fragments of the host gut with the attached gregarines were fixed with 2.5% (v/v) glutaraldehyde in 0.05 M cacodylate buffer (pH 7.4) containing 1.28% (w/v) NaCl in an ice bath in the dark. The fixative was once replaced with fresh fixative after 1 h, and the total fixation time was 2 h. The fixed samples were rinsed three times with cacodylate buffer and post-fixed with 2% (w/v) OsO4 in the cacodylate buffer (ice bath, 2 h).

For SEM study, the fixed gregarines Ancora sagittata were dehydrated in a graded series of ethanol, transferred to an ethanol/acetone mixture (1:1, v/v), rinsed three times with 100% acetone, and critical point-dried with CO2. The samples were mounted on stubs, sputter-coated with gold/palladium, and examined under a CamScan-S2 scanning electron microscope (CamScan, Cambridge, UK).

For TEM study, after dehydration in a graded series of ethanol, the fixed parasites Ancora sagittata were transferred to an ethanol/acetone mixture 1:1 (v/v), rinsed twice in pure acetone, and embedded in Epon resin using a standard procedure. Ultrathin sections obtained using LKB-III (LKB-produkter, Bromma, Sweden) or Leica EM UC6 (Leica Microsystems, Wetzlar, Germany) ultramicrotomes were contrasted with uranyl acetate and lead citrate (Reynolds, 1963) and examined under a JEM-100B or a JEM 1011 electron microscope (JEOL, Tokyo, Japan).

DNA isolation, PCR, cloning, and sequencing

After thrice-repeated rinsing with filtered seawater, the gregarine trophozoites Ancora sagittata were deposited into 1.5 ml microcentrifuge tubes: ∼20 individuals from Roscoff (10 hosts, 2009), ∼20 individuals from WSBS (four hosts, 2006), ∼40 individuals from WSBS (all from the single host, 2010), and ∼100 individuals from WSBS (10 hosts, 2011). All four samples were fixed and stored in “RNAlater” reagent (Life Technologies, Carlsbad, CA, USA).

The nucleotide sequences of Ancora sagittata (SSU, 5.8S, and LSU rDNAs), as well as internal transcribed spacers 1 and 2 (ITS1 and ITS2, respectively) were obtained by two methods: (i) PCR followed by Sanger sequencing (Roscoff and WSBS 2006 samples) and (ii) a genome amplification approach (WSBS 2010 and 2011 samples).

For the first method, DNA extraction was performed using the “Diatom DNA Prep 200” kit (Isogene Laboratory, Moscow, Russia). The rDNA sequences were amplified in several PCRs with different pairs of primers (Fig. 1 and Table 1). As revealed later, the sample WSBS 2006 was contaminated with hyperparasitic microsporidians (Mikhailov, Simdyanov & Aleoshin, 2017), which predominantly reacted with the primers A and B; therefore, a specific forward primer Q5A (Table 1) was constructed to provide a specific gregarine PCR product. A set of overlapping fragments encompassing SSU rDNA, ITS1 and ITS2, 5.8S rDNA, and LSU rDNA was obtained for each sample: fragments I–IV for the sample from Roscoff and fragments V and VI for the sample from the White Sea (Fig. 1). All fragments were amplified with an Encyclo PCR kit (Evrogen, Moscow, Russia) in a total volume of 25 μl using a DNA Engine Dyad thermocycler (Bio-Rad Laboratories, Hercules, CA, USA) and the following protocol: initial denaturation at 95 °C for 3 min; 40 cycles of 95 °C for 30 s, 45 °C (fragments I, II, and V) or 53 °C (fragments II, IV, and VI) for 30 s, and 72 °C for 1.5 min; and a final extension at 72 °C for 10 min. Only weak bands of the expected size were obtained by electrophoresis in agarose gel for fragments II and IV. Therefore, small pieces of the gel were sampled from those bands (using pipette tips on a trans-illuminator), followed by re-amplification with the same primers, “ColorTaq PCR kit” (Syntol, Moscow, Russia), a DNA Engine Dyad thermocycler (Bio-Rad Laboratories, Hercules, CA, USA), and the following PCR conditions: initial denaturation at 95 °C for 1 min; 25 cycles of 95 °C for 30 s, 53 °C for 30 s, and 72 °C for 1.5 min; and a final extension at 72 °C for 10 min. PCR products of the expected size were gel-isolated by the Cytokine DNA isolation kit (Cytokine, St. Petersburg, Russia). For the fragments I, IV, and V, the PCR products were sequenced directly. The fragments II, III, and VI were cloned by using the InsTAclone PCR Cloning Kit (Fermentas, Vilnius, Lithuania) because the corresponding PCR products were heterogeneous. Sequences were obtained by using the ABI PRISM BigDye Terminator v3.1 reagent kit and the Applied Biosystems 3730 DNA Analyzer (Applied Biosystems, Waltham, MA, USA) for automatic sequencing. The contiguous sequences of the ribosomal operons (SSU rDNA + ITS1 + 5.8S rDNA + ITS2 + LSU rDNA) were assembled for the gregarine samples from Roscoff and WSBS 2006 (GenBank accession numbers KX982501 and KX982502, respectively).

Figure 1 Strategy used to obtain contigs of the ribosomal operon from the samples of A. sagittata Roscoff 2009 and A. sagittata WSBS 2006.

Upper part: schematic ribosomal operon with approximate positions of the forward and reverse primers. Lower part: the amplified fragments of ribosomal DNA aligned with the ribosomal operon (above). Numbers indicate the length of the overlapping regions. Roman numerals denote the amplified fragments.

Table 1 Main characteristics of the sequences obtained in this study.

Sample name, obtained resulting sequence and its accession number	Characteristics of the PCR-amplified fragments or NGS-obtained contigs	Method of sequencing and PCR primers (if applicable): forward (F) and reverse (R)	
Ancora sagittata from Roscoff 2009, contig of 4,853 bp long: part of SSU rDNA (1,713 bp), complete ITS1, complete 5.8S rDNA, complete ITS2, and part of LSU rDNA (2,767 bp); KX982501	(I) SSU rDNA (part); 1,567 bp	Sanger (direct sequencing of the PCR product) A1 (F) 5′-GTATCTGGTTGATCCTGCCAGT-3′ r71 (R) 5′-GCGACGGGCGGTGTGTAC-3′	
(II) SSU rDNA (part), ITS1, 5.8S rDNA, ITS2, and LSU rDNA (part); 941 bp	Sanger (after cloning) d6 (F) 5′-CCGTTCTTAGTTGGTGG-3′ 28r32 (R) 5′-CCTTGGTCCGTGTTTCAAGAC-3′	
(III) LSU rDNA (part); 1,748 bp	Sanger (after cloning) 28d12 (F) 5′-ACCCGCTGAAYTTAAGCATAT-3′ 28r72 (R) 5′-GCCAATCCTTWTCCCGAAGTTAC-3′	
(IV) LSU rDNA (part); 1,608 bp	Sanger (direct sequencing of the PCR product) 28d52 (F) 5′-CCGCTAAGGAGTGTGTAACAAC-3′ 28r112 (R) 5′-GTCTAAACCCAGCTCACGTTCCCT-3′	
Ancora sagittata from WSBS 2006, contig of 2,634 bp long: part of SSU rDNA (1,696 bp), complete ITS1, complete 5.8S rDNA, complete ITS2, and part of LSU rDNA (first 564 bp); KX982502	(V) SSU rDNA (part); 1,663 bp	Sanger (direct sequencing of the PCR product) Q5A (F) 5′-GATTAAGCCATGCATGTCT-3′ B1 (R) 5′-GATCCTTCTGCAGGTTCACCTAC-3′	
(VI) SSU rDNA (part), ITS1, 5.8S rDNA, ITS2, and LSU rDNA (part); 1,096 bp	Sanger (after cloning) d71 (F) 5′-GTCCCTGCCCTTTGTACACACCGCCCG-3′ 28r32 (R) 5′-CCTTGGTCCGTGTTTCAAGAC-3′	
Ancora sagittata from WSBS 2010, contig of complete ribosomal operon (5,973 bp) KX982504	Complete sequences of SSU rDNA (1,737 bp), ITS1, 5.8S, rDNA, ITS2, LSU rDNA (3,169 bp), and parts of ETSs	NGS (Illumina HiSeq 2000)	
Ancora sagittata from WSBS 2011 contig of complete ribosomal operon (5,973 bp); KX982503	Complete sequences of SSU rDNA (1,737 bp), ITS1, 5.8S, rDNA, ITS2, LSU rDNA (3,170 bp), and parts of ETSs	NGS (Illumina HiSeq 2000)	
Stentor coeruleus, contig of 3,064 bp long: LSU rDNA, partial sequence; KX982500	LSU rDNA (part); 1,728 bp	Sanger (direct sequencing of the PCR product) 28d12 (F) 5′-ACCCGCTGAAYTTAAGCATAT-3′ 28r72 (R) 5′-GCCAATCCTTWTCCCGAAGTTAC-3′	
	LSU rDNA (part); 1,958 bp	Sanger (after cloning) 28d52 (F) 5′-CCGCTAAGGAGTGTGTAACAAC-3′ 28r132 (R) 5′-DYWRGCYGCGTTCTTCATCG-3′	
Notes:

1 The primer sequences were based on: Medlin et al. (1988).

2 The primer sequences were based on: Van der Auwera, Chapelle & De Wachter (1994).

For the samples of Ancora sagittata 2010 and 2011 (WSBS), DNA extraction was performed using the “NucleoSpin Tissue” kit (Macherey-Nagel, D�ren, Germany). The corresponding complete ribosomal operon sequences were obtained by whole genome amplification and high-throughput sequencing: ∼1 ng of DNA from each sample was amplified with the REPLI-g Midi kit (Qiagen, Hilden, Germany) according to the manufacturer’s protocol and sequenced on an Illumina HiSeq2000 NGS platform (Illumina, San Diego, CA, USA) using one quarter of a lane in paired-end libraries, an estimated mean insert size of ∼330 bp and a read length of 100 bp. The Illumina reads were adapter-trimmed with Trimmomatic-0.30 (Lohse et al., 2012), read pairs with reads shorter than 55 bp were discarded and the remaining reads were assembled using SPAdes 2.5.0 (Bankevich et al., 2012) in single-cell mode (–sc) with read error correction and five k-mer values: 21, 33, 55, 77, and 95. Contiguous sequences corresponding to the ribosomal operon of Ancora sagittata (GenBank accession numbers KX982503 and KX982504) were identified in the assembly using the standalone BLAST 2.2.25+ package (Altschul et al., 1997).

In addition, we sequenced the LSU rDNA of the ciliate Stentor coeruleus to enhance the taxon sampling of the LSU rDNA (GenBank accession number KX982500). Two overlapping sequences was obtained using the same procedures as for the LSU rDNA fragments III and IV of Ancora sagittata (PCR, cloning, and sequencing; see above for details and Table 1 for primers), and the resulting contiguous sequence was assembled.

Predicted secondary structures of ITS2

The structures were created by using MFOLD (Zuker, 2003) under default parameters in the temperature 5–37 °C; it was made manually because there was no suitable template for automatic modelling in the available databases. ITS2 is a genetic region that could be a valuable marker for species delineation and compensatory base changes (CBSs) within it can be used to discriminate species (Coleman, 2000, 2009; Müller et al., 2007; Wolf et al., 2013).

Molecular phylogenetic analyses

Four nucleotide alignments were prepared: of SSU rDNA (two variants: 114 and 52 sequences), LSU rDNA, and ribosomal operon (concatenated SSU, 5.8S, and LSU rDNA sequences). The alignments were generated in MUSCLE 3.6 (Edgar, 2004) and manually adjusted with BioEdit 7.0.9.0 (Hall, 1999): gaps, columns containing few nucleotides, and hypervariable regions were removed. The taxon sampling was designed as to maximalize the phylogenetic diversity and completeness of sequences in the alignments. Representatives of heterokonts and rhizarians were used as outgroups. The final analysis included 114 representative sequences (1,570 aligned sites).

To analyse sequences that are closely related to Ancora sagittata, including environmental entries (from GenBank), we prepared the SSU rDNA alignment of 52 sequences for 1,709 sites; only one sequence was included for each of four clusters of near-identical environmental clones (see below). This analysis involved 139 additional nucleotides from hypervariable regions of the SSU rDNA compared to the standard analysis. To assess the similarity of these closely related sequences quantitatively, substitutions and indels were counted between each pair of the sequences in their overlapping regions and their similarity indexes were calculated as ratio of matching sites to the total amount of sites in the region of overlap, expressed with percentage: ((a − d)/a) × 100%, where a = total number of sites in the region of overlap, d = number of mismatches (Tables S1 and S2).

For the LSU rDNA and ribosomal operon (concatenated SSU, 5.8S, and LSU rDNAs) analyses, the taxon sampling of only 50 sequences was used due to the limited availability of data for LSU rDNA, and, especially, 5.8S rDNA. Therefore, the 5.8S rDNA (155 sites in the alignment) was rejected from the analysis of the concatenated rRNA genes for seven sequences (Chromera velia, Colponema vietnamica, Goussia desseri, Stentor coeruleus, and three environmental sequences: Ma131 1A38, Ma131 1A45, and Ma131 1A49): these nucleotide sites were replaced with “N” in the alignment. The resulting multiple alignments contained 50 sequences (2,911 sites) for the LSU rDNA, and the same 50 sequences (4,636 sites) for the concatenated rDNAs (ribosomal operon). Thus, both taxon sampling comprised an identical set of species, all of which were also represented in the alignment of the 114 SSU rDNA sequences.

Maximum-likelihood (ML) analyses were performed by using RAxML 7.2.8 (Stamatakis, 2006) under the GTR+Г+I model with eight categories of discrete gamma distribution. The procedure included 100 independent runs of the ML analysis and 1,000 replicates of multiparametric bootstrap. Bayesian inference (BI) analyses were computed in MrBayes 3.2.1 (Ronquist et al., 2012) under the same model. The program was set to operate using the following parameters: nst = 6, ngammacat = 12, rates = invgamma, covarion = yes; parameters of Metropolis Coupling Marcov Chains Monte Carlo (mcmc): nchains = 4, nruns = 2, temp = 0.2, ngen = 7,000,000, samplefreq = 1,000, burninfrac = 0.5 (the first 50% of the 7,000 sampled trees, i.e., the first 3,500, were discarded in each run). The following average standard deviations of split frequencies were obtained: 0.009904 for the SSU rDNA analysis, 0.001084 for the LSU rDNA analysis, and 0.001113 for the ribosomal operon analysis. The calculations of bootstrap support for the resulting Bayesian trees were performed by using RAxML 7.2.8 under the same parameters as for the ML analyses (see above).

Results

Light and SEM

Thirty specimens of Capitella capitata from Roscoff (English Channel, France) and 25 specimens from the WSBS (Russia) were dissected. All were infected and the number of gregarine trophozoites per host varied from several individuals up to about a 100. The parasites from both locations had the same morphology, which fitted the description of Ancora sagittata: an elongated body that narrowed toward the posterior end and with a rounded anterior end, without a septum, and with two lateral projections giving the cell the appearance of an anchor (Perkins et al., 2000). The average dimensions were 250 μm in length and 37 μm in width (n = 25). The attached trophozoites were easy to dislodge from the host epithelium, and a small drop of the cytoplasm then appeared at the front of the gregarine. However, most of the gregarines were already free (not attached) during the dissection, without any visible damage to their forebodies. All detached gregarines demonstrated gliding motility in seawater. Other stages of the life cycle were not observed (Fig. 2).

Figure 2 Light (A) and scanning electron microscopy (B–G) of Ancora sagittata.

(A and B) General view of the gregarine; (C) Epicyte; (D) View of the gregarine from the apical pole of the cell; (E) Epicytic folds at the base of the lateral projections (lp); (F, G) Apical pole of Ancora sagittata (arrows) with (F) and without the apical papilla (G). lp, lateral projections of the cell.

SEM micrographs of the gregarine surface revealed the structure typical for eugregarines (Vávra & Small, 1969): epicytic folds appressed to each other (threefold per 1 μm) and converging to the apex of the cell, where a small apical papilla of 2.5 μm in diameter was sometimes observed (Figs. 2F and 2G). The epicytic folds branched dichotomically in the apical region and similar branching was also observed at the bases of the lateral projections (Fig. 2E).

Transmission electron microscopy

Cross-sections of trophozoites of Ancora sagittata showed a typical eugregarine tegument (Vivier, 1968; Vivier et al., 1970; Schrével et al., 1983): the epicyte consisting of numerous folds or crests (Figs. 3A–3F) formed by the trimembrane pellicle (45 nm thick) composed of the plasma membrane (covered by the cell coat) and the inner membrane complex (IMC) (Fig. 3D). Cross-sections of the middle of the cell revealed regularly arranged and closely packed epicytic folds (crests) that were approximately 1 μm high and 375 nm wide and had finger-like shapes with weak constrictions at their bases. A 13 nm thick internal lamina (an electron-dense layer undelaying the pellicle) was observed, which was thickened at the bottom of grooves between the epicytic folds (up to 48–50 nm) and did not form links in the bases of the folds, which are characteristic for many other eugregarines (see Vivier, 1968; Schrével et al., 2013; see Discussion). Six to eight rippled dense structures (also called apical arcs) were present at the top of the folds. No 12 nm apical filaments, characteristic for most eugregarines (Vivier, 1968; Schrével et al., 2013), were detected, but electron-dense plates were found at the top of the folds just beneath the IMC (Fig. 3D). The cytoplasm in the folds contained fibrils (Fig. 3C). The folds displayed increased bulging near the bases of the lateral projections of the trophozoite cell, and rare micropores were observed at the lateral surfaces of the folds in this region (Fig. 3F). In the frontal region of the cell, gaps in between the folds were present and large electron-dense globules were found in the cytoplasm of the folds (Fig. 3E). Circular filaments (∼30 nm) were observed just beneath the tegument (Figs. 3C, 3F and 3G). The ecto- and endoplasm were not separated distinctly from each other (Figs. 3A and 3B): the thickness of the ectoplasm (the cytoplasm layer free of amylopectin) varied only from 1.5 to 2.5 μm. Rounded amylopectin granules of approximately 0.7–1 μm were abundant in the deeper layers of the cytoplasm (Figs. 3A and 3B).

Figure 3 Transmission electron microscopy of Ancora sagittata.

(A–C) Cross sections in the middle of the cell show epicytic folds (ef) with fibrils (f) inside, internal lamina (il), circular cortical filaments (cf), and granules of amylopectin (ap); (D) Cross section through the top of the epicytic fold reveals a structure of the pellicle consisting of the plasma membrane (pm) and the internal membrane complex (imc) with rippled dense structures (= apical arcs, aa) and an electron dense plate (arrow); (E) Cross section through the epicytic folds of the frontal zone of the cell with electron dense globules inside the folds; (F) Cross section at the level of the lateral projections: a micropore (mp) and circular cortical filaments (cf) are visible; (G) Tangential section of the cortex in the posterior region of the trophozoite reveals circular cortical filaments (cf).

Figure 4 Transmission electron microscopy of the attachment apparatus of Ancora sagittata.

(A) Longitudinal section of the gregarine forebody embedded in a host cell shows a large frontal vacuole (fv) and amylopectin granules (ap) within the attachment organelle and the main part of the cell; the black arrows indicate the base of the contact zone (circular groove, see B); (B) Longitudinal section through the base of the contact zone between the gregarine and host cell under a higher magnification: gregarine cell forms a circular groove (black arrow) pinching the host cell; the rear wall of the frontal vacuole (double arrow) arises from this area; parallel filaments (fi) arise from the groove zone backward; the white arrow indicates the terminus of the internal membrane complex (imc) of the pellicle; pm is the plasma membrane of the gregarine cell.

Figure 5 Diagram of the contact between the gregarine and the host cell as inferred from TEM micrographs.

Abbreviations are the same as in Fig. 4.

The trophozoites were attached to the intestinal epithelium by a bulbous attachment apparatus that was embedded in the host cell and connected to the trophozoite cell by a short stalk or neck (Fig. 4A). The anterior part of the attachment apparatus contained a large lobate vacuole filled with a loose, thin fibrillar network, which could be the result of coagulation of some matter during the fixation and/or embedding procedures (Figs. 4A and 4B). A groove pinching a small portion of the host cell was present at the base of the attachment bulb (Figs. 4B and 5). The IMC of the parasite pellicle terminated at this site and the attachment bulb was apparently covered only by the single plasma membrane of the gregarine, not by the pellicle (Figs. 4B and 5). A bundle of longitudinal filaments spread throughout the gregarine cell backwards from the IMC terminus. The wall of a large frontal vacuole arose from the same site (Figs. 4B and 5). The cytoplasm behind the vacuole contained individual amylopectin granules (Fig. 4A). Longitudinal sections of four trophozoites revealed the complex structure of the contact zone between the parasite and the host cell (Figs. 4B and 5). The cell junction was formed by two closely adjacent plasma membranes of the host and parasite cells without a distinct gap between them. Electron-dense areas were present on both parasite and host cell sides of the junction. The area within the host cell appeared uniformly gray, whereas that of the gregarine cell was distinguished into three zones: (i) a black zone immediately adjacent to the cell junction, (ii) a gray zone similar to that in the host cell, and (iii) thin fibrils arising from the gray zone toward the interior of the cell (Figs. 4B and 5).

Sequence diversity in Ancora sagittata

Four contiguous nucleotide sequences of Ancora sagittata were obtained (Table 1), three (Roscoff, WSBS 2010, and 2011) covering complete or near-complete ribosomal operon (SSU, 5.8S, LSU rDNAs, and the internal transcribed spacers ITS1 and ITS2), and a shorter one (WSBS 2006) lacking the most part of LSU rDNA (only first ∼600 bp of it were amplified and sequenced; Fig. 1). Three of these sequences (Roscoff, WSBS 2006, and 2011; ribotype 1) were near identical to one another (99.4%–100%; Tables S1 and S2), whereas the fourth (WSBS 2010; ribotype 2) was more divergent (94.3%–96.2% identities with three other sequences). Nucleotide substitutions and indels were concentrated chiefly in the hypervariable regions of the rRNA genes and in the ITSs (the ITSs contained ∼40% of total mismatches).

A search for CBCs in ITS2 was performed to discriminate possible cryptic species (Coleman, 2000, 2009; Müller et al., 2007; Wolf et al., 2013). The manually assembled secondary structure was tested by MFOLD in the temperature of 5–37 °C and was found to be nearly optimal. The Ancora sagittata ITS2 (Fig. 6) appeared to be one of the shortest known sequences in eukaryotes (102 and 100 nucleotides in the ribotype 1 and 2, respectively; helix IV was absent); however, it retained universally conserved features (Schultz et al., 2005): a U–U mismatch in helix II and a vestige of the “UGGU” motif in helix III, modified as “UGUGU” (Fig. 6). Four CBCs between the ribotypes 1 and 2 were detected: two putative in the basal helix, one in helix I, and one at the base of helix III.

Figure 6 Predicted secondary structures of ITS2 transcripts of two Ancora sagittata ribotypes demonstrating differences between them.

(A) Ribotype 1; (B) Ribotype 2. Nucleotide substitutions and insertions in the ribotype 2 are highlighted in gray. Nucleotides involved in compensatory base changes are encircled.

Phylogenies inferred from SSU rDNA

Phylogenies of SSU rDNA (114 sequences; 1,570 sites) showed a well supported monophyly of the major groups of alveolates (ciliates, dinoflagellates and their subgroups, and apicomplexans) with a high Bayesian posterior probability (PP) and moderate ML bootstrap percentage (BP) support (Fig. 7). The backbone of the apicomplexans was poorly resolved in both Bayesian and ML analyses; nevertheless, the topologies were largely congruent with small differences in the gregarine branching order. The cryptosporidians were consistently placed as the sister group of all gregarines in both analyses, although with low support. Archigregarines (Selenidium spp.) formed three branches of greatly variable lengths and were not monophyletic. Eugregarines were separated from archigregarines and were monophyletic in both Bayesian and ML trees, although without a cogent support (PP = 0.58, BP = 12%). They comprised eight well-supported subclades of an uncertain branching order (Fig. 7), five of which were recently erected as superfamilies (Clopton, 2009; Rueckert et al., 2011; Simdyanov & Diakin, 2013; Cavalier-Smith, 2014), namely: (i) Lecudinoidea (Veloxidium leptosynaptae and the aseptate marine Lecudinidae (with the type species Lecudina pellucida) and Urosporidae); (ii) Cephaloidophoroidea (septate and aseptate gregarines from crustaceans); (iii) Gregarinoidea (septate gregarines from insects); (iv) Stylocephaloidea (septate gregarines from insects); and (v) Actinocephaloidea (septate and some aseptate gregarines from insects including neogregarines and Monocystis agilis). Two additional lineages were designated as incertae sedis: one (vi) was composed entirely of unidentified environmental sequences including a “clone from the foraminiferan Ammonia beccarii” that actually is an apicomplexan sequence (Pawlowski et al., 1996), and the other (vii) comprised the aseptate marine gregarine Paralecudina polymorpha and related environmental sequences. The last lineage (viii), hereby named “Ancoroidea,” was a robust monophyletic clade that includes Ancora sagittata, Polyplicarium spp., and 70 environmental sequences from anoxic marine habitats (Figs. 7 and 8). Two clusters were found within this clade (Fig. 8): a robust clade including Ancora sagittata and related environmental sequences, and the clade of Polyplicarium spp. and environmental relatives, which was either strongly (Fig. 7) or moderately (Fig. 8) supported depending on the dataset. The environmental clade vii (“clone from Ammonia beccarii” and relatives) displayed a certain affinity to the Ancoroidea (Figs. 7 and 8).

Figure 7 Bayesian inference tree of alveolates calculated by using the GTR+Г+I model from the dataset of 114 SSU rDNA sequences (1,570 sites).

Numbers at the nodes indicate Bayesian posterior probabilities/ML bootstrap percentage. Black dots on the branches indicate Bayesian posterior probabilities and bootstrap percentages of at least 0.95% and 95%, respectively. The newly obtained sequences of Ancora sagittata are highlighted in by a black rectangle. Asterisks indicate aseptate gregarines within the “septate” clade; arrows indicate neogregarines.

Figure 8 Bayesian inference tree of Ancora sagittata and related sequences obtained by using the GTR+Г+I model from the dataset of 52 SSU rDNA sequences (1,709 sites).

Numbers at the nodes indicate Bayesian posterior probabilities/ML bootstrap percentage. Black dots on the branches indicate Bayesian posterior probabilities and bootstrap percentages of at least 0.95% and 95%, respectively. The newly obtained sequences of Ancora sagittata are highlighted by black rectangles. Black triangles indicate clusters of near-identical sequences (identity of 99% or more), each of which was represented by a single representative.

Aseptate gregarines (Ancoroidea, Lecudinoidea, and the clade of Paralecudina) were not monophyletic, whereas the four other lineages (Actinocephaloidea, Cephaloidophoroidea, Gregarinoidea, and Stylocephaloidea) formed a weakly supported clade of primarily septate eugregarines, although some representatives of this “septate” clade are actually aseptate (marked with asterisks in Fig. 7).

Analyses of LSU rDNA and the ribosomal operon

All analyses of the LSU rDNA dataset (50 sequences, 2,911 sites) showed topologies that were congruent with the SSU rDNA result with minor exceptions (Fig. 9A). The three sequences from Ancora sagittata (Roscoff, WSBS 2010, and 2011) were monophyletic and related to a clade containing Gregarina spp. and crustacean gregarines in both Bayesian and ML analyses (Fig. 9A). All gregarines, including a clade containing Ascogregarina and Neogregarinida sp. OPPPC1 (Fig. 9A), were monophyletic.

Figure 9 Bayesian inference trees of the alveolates obtained by using the GTR+Г+I model and 50 sequences.

(A) LSU rDNA dataset (2,911 sites); (B) Ribosomal operon dataset (4,636 sites). Numbers at the nodes indicate Bayesian posterior probabilities/ML bootstrap percentages. Black dots on the branches indicate Bayesian posterior probabilities and bootstrap percentages of at least 95% and 95%, respectively. The newly obtained sequences of Ancora sagittata are highlighted by black rectangles. Accession numbers in (B) are arranged in following order: SSU rDNA, 5.8S (if available), LSU rDNA. The sequences of Babesia bigemina were obtained from the Sanger Institute genome project (https://www.sanger.ac.uk/Projects/B_bigemina/). Asterisks mark partial LSU rDNA sequences of small size (300–700 bp).

Trees derived from the ribosomal operon dataset (alignment of 50 sequences, 4,636 sites) showed the same topology as the LSU rDNA tree (Fig. 9B) with increased supports for several branches. Within the sporozoan clade, all gregarines were monophyletic and most supports were similar to those in the LSU rDNA tree; however, BP support for the subclade including Ancora sagittata from Roscoff and Ancora sagittata from WSBS 2011 increases considerably (BP = 92% vs. 86% in the LSU rDNA tree).

Discussion

Ultrastructure of the cortex

The tegument of gregarines is composed of a trimembrane pellicle (plasma membrane and IMC, formed by two closely adjacent cytomembranes) underlaid by the internal lamina, an electron-dense layer that most likely consists of closely packed thin fibrils (Vivier, 1968; Vivier et al., 1970; Schrével et al., 1983, 2013). The pellicle forms so-called epicyte—a set of multiple narrow longitudinal folds called epicytic folds (Vivier, 1968; Vávra & Small, 1969; Vivier et al., 1970; Schrével et al., 1983, 2013). The epicytic folds of most eugregarines, including the Lecudinidae, have a very specific structure: they contain several rippled dense structures (also called apical arcs) and 12 nm apical filaments within their top regions: the former are located between plasmalemma and the IMC, the latter just beneath the IMC; the internal lamina usually forms links or septa in the bases of the folds (Vivier, 1968; Schrével et al., 1983, 2013; Simdyanov, 1995b, 2004, 2009). Both the rippled dense structures and 12 nm apical filaments are thought to be involved in gliding motility, which is characteristic of typical eugregarines; however, the detailed mechanism of gliding remains unclear (Vivier, 1968; Vávra & Small, 1969; Mackenzie & Walker, 1983; King, 1988; Valigurová et al., 2013). Observations of Ancora sagittata trophozoites are congruent with the information available for eugregarines, with a few differences. The 12 nm apical filaments were not observed in the epicytic folds, they are obscured or substituted by an electron dense plate. Additional studies, including immunochemical methods, are required to reveal the true composition of this structure.

Another interesting observation is the absence of the links of the internal lamina (see above) in the bases of the epicytic folds of Ancora sagittata (Figs. 3C and 3F) and, simultaneously, the accumulation of large cytoplasmic inclusions within the folds and the location of micropores on the lateral surfaces of the epicytic folds, while their typical location is in between the folds. These three peculiarities have been reported for a few other eugregarines such as Kamptocephalus mobilis, Mastigorhynchus bradae, and Stylocephalus spp. (Desportes, 1969; Simdyanov, 1995a) and they have occurred together in each case, so they obviously correlate with one another. Thus, in eugregarines with typical epicytes, the links of the internal lamina appear to act as barriers between the space within the epicytic folds and the rest of the cytoplasm, but their true purpose remains unclear.

Circular cortical filaments found in Ancora sagittata are similar to those in eugregarines with peristaltic motility, such as Monocystis spp., Nematocystis magna, and Rhynchocystis pilosa (Miles, 1968; Warner, 1968; Vinckier, 1969; MacMillan, 1973), or bending motility, such as some Gregarina spp. (Valigurová et al., 2013). However, neither peristaltic nor bending motility have been observed in Ancora sagittata.

Attachment apparatus: mucron or epimerite?

The terminology of the gregarine attachment apparatuses is rather confusing. The commonly used names are “mucron” and “epimerite” depending on whether the trophozoite is aseptate or septate, respectively (Levine, 1971). The mucron is mostly small and can be pointed, rounded or sucker-shaped, whereas the epimerite varies in size and shape from elongated to lenticular and is sometimes equipped with hooks or other projections (Grassé, 1953; Schrével et al., 2013). Consequently, the attachment apparatus of Ancora sagittata should be called the mucron (see Perkins et al., 2000), although it more resembles the epimerite on its ultrastructure and fate (see below). Similar confusion arises about the attachment apparatus in other gregarines and stems from Levine’s definition (Levine, 1971) that “[the] mucron is an attachment organelle of aseptate gregarines…,” i.e., it applies equally to archigregarines and aseptate eugregarines as both of them are aseptate. TEM data has since forced a revision of this light microscopy-driven perspective revealing conspicuous differences between the attachment organelles of archi- and eugregarines. The archigregarine mucron contains an apical complex and performs myzocytotic feeding, i.e., intermittent sucking of nutrients through a temporary cytostome (Schrével, 1968, 1971a; Simdyanov & Kuvardina, 2007; Schrével et al., 2016), whereas eugregarine trophozoites have no apical complex (with exception of the earliest developmental stages) and do not exhibit myzocytosis in their mucrons and epimerites (Schrével & Vivier, 1966; Devauchelle, 1968; Baudoin, 1969; Desportes, 1969; Ormierès & Daumal, 1970; Hildebrand, 1976; Ormierès, 1977; Tronchin & Schrével, 1977; Ouassi & Porchet-Henneré, 1978; Valigurová & Koudela, 2005; Valigurová et al., 2007; Valigurová, Michalková & Koudela, 2009; Schrével et al., 2013, 2016). Thus, there is no doubt that the mucron of archigregarines and the epimerite in septate eugregarines differ in their genesis, structure, and feeding function (Table 2 and Fig. 10). However, the “mucron” of aseptate eugregarines (e.g., some lecudinids) is actually a homologue of the epimerite when examined in detail, not of the archigregarine mucron (Table 2 and Fig. 10). The archigregarine mucron contains the apical complex and is covered by a trimembrane pellicle, excepting a small region in front of the conoid where the IMC is absent and the cytostome is intermittently opened (Schrével, 1968, 1971a; Kuvardina & Simdyanov, 2002; Simdyanov & Kuvardina, 2007). In contrast, the attachment organelles of eugregarine trophozoites, both the “mucron” and epimerite, originate from the region corresponding to the “cytostome site” of archigregarines (in front of the conoid where is no IMC) as a progressing protuberance covered by a single plasma membrane (Table 2, Figs. 10F and 10G) and their apical complex disappears in a short time after the protuberance is starting to develop (Desportes, 1969; Tronchin & Schrével, 1977; Ouassi & Porchet-Henneré, 1978). The archigregarine mucron forms a septate cell junction with the host cell, the main characteristic of which is a conspicuous “septate” gap between the host plasma membrane and the gregarine pellicle (Simdyanov & Kuvardina, 2007). In contrast, a cell junction in the eugregarine attachment apparatus (both the so-called “mucron” and epimerite) forms no gap between the parasite and host cell membranes, which are underlaid by electron dense areas (Figs. 5 and 10). The eugregarine cell junction zone is bordered by the circular groove running along the edge of the attachment site and pinching a small portion of the host cell; this structure is absent in archigregarines. In archigregarines, a large mucronal vacuole is present within the mucron, which is intermittently connected by a duct (cytopharynx) with the cytostome; obviously it is a food vacuole (Schrével, 1968, 1971a; Kuvardina & Simdyanov, 2002; Simdyanov & Kuvardina, 2007; Schrével et al., 2016). In eugregarine trophozoites, no cytostome-cytopharyngeal complex has been observed—only with the possibly exception of the earliest developmental stage (Figs. 10F1 and 10G3). A large frontal vacuolar structure (usually flattened and containing fibrillar matter) develops just beneath the cell junction region; however, the vacuole can sometimes be completely replaced with a dense fibrillar zone (Tronchin & Schrével, 1977; Ouassi & Porchet-Henneré, 1978). No evidence of its involvement in the eugregarine feeding has been observed. Finally, archigregarines retain their mucron (together with the apical complex) well into the syzygy stage (Fig. 10D) (Kuvardina & Simdyanov, 2002). Although the fate of the attachment apparatus in aseptate eugregarines is poorly studied, it is presumably the same that in the septate eugregarines, whose gamonts lose their epimerite upon the detaching from the host epithelium (Grassé, 1953; Devauchelle, 1968; Valigurová, Michalková & Koudela, 2009; Schrével et al., 2013).

Table 2 Comparison of the key features of gregarine attachment organelles.

	Mucron of archigregarines (Selenidium)	“Mucron” of aseptate eugregarines	Epimerite of septate eugregarines	
Shape	Knob-like	Sucker-shaped or dome-shaped	Various, usually—a well-developed frontal protuberance of the cell of diverse shape	
Tegument structure in the region of the junction with the host cell	The tegument of mucron is trimembrane pellicle excepting small region in front of conoid, where a cytostome and duct of mucron vacuole is intermittently formed, IMC is absent and there is just a single plasma membrane	The IMC of the pellicle terminates at the edge of the cell junction zone, so the tegument of the attachment organelle is represented only by a single plasma membrane	The IMC of the pellicle terminates at the edge of the cell junction zone, so the tegument of the attachment organelle is represented only by a single plasma membrane	
Cell junction between host and parasite	Septate cell junction; no peculiar structures on the edge of the junction zone	Two closely adjacent plasma membranes (of host and parasite) forming high electron density zone. A circular groove in the gregarine tegument (plasma membrane) runs along the edge of the region of the cell junction (where the IMC terminates) and pinches a small portion of the host cell	Two closely adjacent plasma membranes (of host and parasite) forming high electron density zone. A circular groove in the gregarine tegument (plasma membrane) runs along the edge of the region of the cell junction (where the IMC terminates) and pinches a small portion of the host cell	
Cytoplasm organelles	Apical complex (conoid, apical polar ring(s), rhoptries) and mucronal (food) vacuole	Frontal region of “mucron” contains fibrillar zone or large vacuole with fibrillar content adjoining the cell junction zone; no mitochondria were observed; longitudinal actin-like fibrillar structures (filaments) are well developed	Frontal region of epimerite contains a large flattened frontal vacuole with fibrillar content adjoining the cell junction zone; it is built from ER vesicles; mitochondria, often numerous and arranged in a layer, are located beneath this vacuole during growth and development of the trophozoite; different inclusions (lipid globules, amylopectin granules); longitudinal fibrillar structures (microfilaments and microtubules) can be well developed within the stalk of epimerite (if present)	
Functioning	Attachment and feeding by myzocytosis: formation of temporary cytostome-cytopharingeal complex consisting of mucronal vacuole with the duct running through the conoid	Attachment; feeding is questionable, no myzocytosis observed	Attachment; feeding is questionable, no myzocytosis observed	
Fate	Mucron of archigregarines persists for long time after trophozoite detachment and retains apical complex (conoid at least) till (including) stage of syzygy	Unknown; most likely is to retract/condense: frontal part of mature detached gamonts is covered by trimembrane pellicle, but not by a single plasma membrane	When trophozoite transforms into mature gamont, the epimerite is to break off or to retract/condense	
Studied species	Selenidium pendula, S. hollandei, S. orientale, S. pennatum	Lecudina sp. from Cirriformia tentaculata, L. pellucida, Lankesteria levinei, Difficilina cerebratuli	Didymophyes gigantea, Epicavus araeoceri, Gregarina spp., Leidyana ephestiae, Pyxinia firmus, Stylocephalus africanus	
References	Schrével (1968), Schrével (1971a), Kuvardina & Simdyanov (2002), Simdyanov & Kuvardina (2007), Schrével et al. (2016)	Schrével & Vivier (1966), Ouassi & Porchet-Henneré (1978), Simdyanov (1995b), Simdyanov (2009)	Grassé (1953), Devauchelle (1968), Baudoin (1969), Desportes (1969), Ormierès & Daumal (1970), Hildebrand (1976), Ormierès (1977), Tronchin & Schrével (1977), Marquès (1979), Ghazali et al. (1989), Valigurová & Koudela (2005), Valigurová et al. (2007), Valigurová, Michalková & Koudela (2009), Schrével et al. (2013)	

Figure 10 Comparison of the attachment organelles of archigregarines Selenidium spp. (A–E) with septate and aseptate eugregarines (F–I).

(A) Drawing of the apical part of a Selenidium hollandei cell; (B) Ultrastructure of the apical part of an S. orientale cell, a longitudinal section; (C) The frontal region of the mucron under a higher magnification; (D) Mucron of the gamont (syzygy partner) of S. pennatum, a longitudinal section; (E) Predicted myzocytotic feeding in Selenidium; the mucron is embedded in the host cell and contains well-developed apical complex consisting of the conoid (co), polar ring (pr) giving rise to subpellicular microtubules (smt), rhoptries (rh) with rhoptry ducts (rd), and a large mucronal vacuole (mv); the tegument of the mucron comprises a trimembrane pellicle (pe) consisting of the plasma membrane (pm) and internal membrane complex, IMC (imc), with the exception of a small region in front of the conoid, a “cytostome site,” where the IMC is absent and only single plasma membrane is present; the cytostome is intermittently opened in this region to myzocytosis: at first, food comes through the duct (temporary cytopharynx) in the newly formed mucronal vacuole (mv), which then becomes a food vacuole (fv) and is transported into the cell along microtubules (mt) for digestion; the parasite-host contact is mediated by the septate cell junction (scj) with a characteristic wide gap between the plasma membranes (pm and hm, respectively). The mucron with the apical complex persists for a long time into the syzygy; the mucronal food vacuole is absent because the syzygy is a non-feeding stage (D). (F) Development of trophozoite of the septate gregarine Gregarina blaberae (scheme): (i), epimerite (ep) develops as a bulb in front of the apical complex consisting of the conoid and axial organelle (ao), which is likely a homologue of mucronal vacuole (also see (Giii)); the IMC terminates near the apical part of conoid (similarly to mature Selenidium), therefore the developing epimerite is covered only by a single plasma membrane, not by the pellicle; (ii–vi), the apical complex disappears, the epimerite is growing; a large flattened frontal vacuole (frv) arising from the layer of membrane alveoli (ma) of endoplasmic-reticulum (er) origin, numerous mitochondria (m), granules of storage carbohydrate amylopectin (sc), lipid drops (ld), and vacuoles (v) are present in the epimerite cytoplasm; (vi), finally, protomerite (p) and deutomerite (d) are separated by the septum (s). (G) Comparison of developing attachment organelles in the youngest trophozoites of the aseptate gregarine Lecudina sp. from the polychaete Cirriformia (Syn. Audouinia) tentaculata: ((i) and (ii); (ii) shows the details of the cell junction marked by the rectangle in (i)) and G. blaberae ((iii), the magnified fragment of (Fi)): both organelles develop ahead of the conoid in the same way and are covered by a single plasma membrane; the cell junction (cj) between the parasite and host cells is, unlike Selenidium, formed by two closely adjacent plasma membranes (parasite and host); an electron-dense fibrillar zone adjoins the cell junction in the gregarine cell (arrow); the cell junction is bordered by the circular groove (cg) pinching a small portion of the host cell; the IMC terminates (it) at the apical part of the conoid. (H) Comparison of the “mucron” of a well-developed trophozoite of the same Lecudina sp. ((i) and (ii); (ii) is the magnified fragment of (i) marked by the rectangle) and underdeveloped epimerite (ep) of a growing trophozoite of G. blaberae (iii), the same stage as in (Fiv): the IMC terminates (ie) at the base of the attachment organelle (it marks the former apex of the sporozoite mucron), the cell junction consists of two closely adjacent plasma membranes bordered by the circular groove (cg) pinching a small portion of the host cell, a large flattened frontal vacuole (frv) with fibrillar content develops just beneath the region of cell junction. (I) Comparison of the developing epimerite of an older trophozoite of G. blaberae ((i), stage (vi) from (F), magnified) and the attachment organelle of Lecudina (Syn. Cygnicollum) lankesteri (ii); (m), mitochondria. (J) A trophozoite and mature gamonts of L. lankesteri: losing of the epimerite. (A) is reprinted from: Schrével, 1968 (© 1968 Société Française de Microscopie Electronique, Paris), with permission from the Journal de Microscopie et Biology Cellulaire published by Société Française de Microscopie Electronique, Paris; (B, C, and E) are reprinted from: Simdyanov & Kuvardina, 2007 (© 2007 Elsevier), with permission from Elsevier (D) is reprinted from: Kuvardina & Simdyanov, 2002 (© 2002 by Russia, Protistology), with permission from the journal Protistology (Apr 19, 2017); (F, Giii, Hiii, and Ii) are reprinted from: Tronchin & Schrével, 1977 (© 1977 Society of Protozoologists, © John Wiley and Sons), with permission from John Wiley and Sons (Gi, Gii, and Hi) are reprinted from: Ouassi & Porchet-Henneré (1978), with permission from Elsevier #RP016388; (Iii and J) are reprinted from: Desportes & Théodoridès, 1986 (© 1986 Elsevier), with permission from Elsevier #RP016388.

The structure and cytoplasmic content of developed epimerites in septate gregarines vary substantially (Devauchelle, 1968; Baudoin, 1969; Desportes, 1969; Ormierès & Daumal, 1970; Tronchin & Schrével, 1977; Valigurová & Koudela, 2005): numerous mitochondria, granules of amylopectin, lipid drops and vacuoles, well developed arrays of ER, and numerous fibrillar structures (microtubules and microfilaments)—especially in the basal region if it is shaped like a neck or stalk (so-called “diamerite”) as, e.g., in Epicavus araeoceri (Ormierès & Daumal, 1970), can be present.

The attachment apparatus of Ancora sagittata displays the main features of a simple epimerite (Fig. 4) lacking cytoplasmic organelles and inclusions (absence of mitochondria, ER, and lipid drops), although, like in epimerites of septate gregarines, there are amylopectin granules and a large frontal vacuole—although not flattened, but rather bulky. Detached gregarines (mature gamonts?) have no epimerite, which has been apparently discarded, judging from the appearance and behavior of individuals that were artificially dislodged from the host epithelium (see Results). Some other aseptate gregarines also possess complex attachment organelles that are comparable to true epimerites of septate gregarines in shape, ultrastructure, and fate (absent in mature gamots), e.g., Lecudina (Cygnicollum) lankesteri. Unlike Ancora sagittata, the epimerite of L. lankesteri (Figs. 10I and 10J) has a complex structure: the cytoplasm contains mitochondria, inclusions, a well-developed cytoskeleton, and abundant fibrillar structures in the basal region (Desportes & Théodoridès, 1986). Thus the sucker-shaped “mucrons” of many other lecudinids, such as Lecudina sp. from the polychaete Cirriformia (Audouinia) tentaculata (Ouassi & Porchet-Henneré, 1978; also see Figs. 10G and 10H) and L. pellucida (Schrével & Vivier, 1966), which are not deeply embedded into host cell, can be considered underdeveloped epimerites that lack the main (middle) region containing cytoplasmic organelles and inclusions.

Taking into account the homologies of the eugregarine attachment organelles, the term “mucron” should be restricted to the attachment apparatus in archigregarines, which contains the apical complex and performs myzocytosis. In eugregarines, both aseptate and septate, the term “epimerite” appears to be more appropriate, and is in accordance with the definitions and gregarine descriptions in the “classic” literature on gregarines (Watson Kamm, 1922). This terminological correction will remove ambiguity in taxonomical diagnoses and emphasize that the epimerite is a shared evolutionary innovation (synapomorphy) of eugregarines. More representative data are required to distinguish different types of epimerites: for example, the “cephaloid” type of Cephaloidophora (Simdyanov, Diakin & Aleoshin, 2015) and the underdeveloped epimerites of Lecudina spp. (above), which can be called “pseudomucrons.” It should be noted, that in some morphologically divergent eugregarines, e.g., Uradiophora maetzi and representatives of the family Dactylophoridae, the epimerite was reduced and they are anchored in host cells with projections of the protomerite (Ormierès & Marquès, 1976; Desportes & Théodoridès, 1985).

Reconciling molecular phylogenies with eugregarine morphology

In SSU rDNA phylogenies published to date, gregarines (sensu class Gregarinomorpha Grassé, 1953) have been not monophyletic. The most probable reason for that is that the SSU rDNA sequences of many gregarines are highly divergent, therefore the topologies of resulting phylogenetic trees are sensitive to changes in alignment site selection and taxon sampling. Additionally, the presence of many long branches among gregarines and other apicomplexans may lead to long branch attraction (LBA) artefacts (Bergsten, 2005). Only after careful manual editing of the alignment (see supplemental raw data) and the exclusion of single gregarine sequences corresponding to three extremely long branches (Pyxinia crystalligera, Stenophora robusta, and Trichotokara spp.), all of the other gregarines did form a monophyletic lineage, albeit weakly supported (Fig. 7). Despite their weakly supported monophyly in the SSU rDNA phylogenies, all gregarines display a distinct morphological synapomorphy: the gametocyst, which is an encysted syzygy (Frolov, 1991). Among other apicomplexans, only adeleid coccidians have syzygy, but without the subsequent encystment into a gametocyst. Unlike SSU rDNA alone, both the LSU rDNA and ribosomal operon-based phylogenies support the monophyly of gregarines, although with a significantly limited taxon sampling without archigregarine and some eugregarine lineages (Fig. 9). Similarly to SSU rDNA phylogenies, LBA can affect these tree topologies, although its negative effects are expected to be lower than in SSU rDNA-alone phylogenies because the relative evolution rates of the LSU rDNA in apicomplexans are more even than those of the SSU rDNA (Simdyanov, Diakin & Aleoshin, 2015).

Neogregarines (order Neogregarinida Grassé, 1953) never form a monophyletic lineage but are shuffled amongst actinocephalids (confirming Grassé’s hypothesis for their origin) and should therefore be included in the superfamily Actinocephaloidea. Consequently, the absence of merogony should be removed from the eugregarine diagnosis. Archigregarines are paraphyletic in SSUr DNA-based phylogenies, forming two or three independent lineages, which are often shuffled with eugregarine clades in available molecular phylogenetic trees (Cavalier-Smith, 2014). However, the proposition of the independent polyphyletic origin of different eugregarine lineages (Cavalier-Smith, 2014) contradicts evidence from ultrastructural studies. On the contrary, relying on the morphological evidence, eugregarines appear to be a monophyletic group because all their major lineages share at least two distinct morphological apomorphies (Fig. 11): (i) the presence of the epimerite (see above) and (ii) gliding motility apparently associated with rippled dense structures (apical arcs) and 12 nm apical filaments in eugregarine epicytic folds (see Krylov & Dobrovolskij, 1980). Archigregarines, apart from having the different type of attachment apparatus (the mucron, see above), lack the eugregarine type of the epicyte: they possess longitudinal pellicular folds (bulges), which are significantly larger than eugregarine epicytic folds (crests) and contain neither the rippled dense structures (apical arcs) nor 12 nm apical filaments (Fig. 11A); just beneath the pellicle, both in the bulges and between them, one to three layers of longitudinal subpellicular microtubules are located that have never observed in the eugregarines (Schrével, 1971a, 1971b; Simdyanov & Kuvardina, 2007; Schrével & Desportes, 2013b). The archigregarine pellicular bulges are thus only a simple surface sculpture, whereas the epicytic crests of eugregarines are complex organelles that apparently provide the gliding motility, which is absent in archigregarines and substituted by active bending motility instead (Grassé, 1953; Schrével, 1971a; Perkins et al., 2000). The failure to distinguish these different cortical structures in archi- and eugregarines (Cavalier-Smith, 2014) is linked to the use of a misleading term “longitudinal folds,” which assumes their identity in both gregarine groups and has been used as the morphological evidence of the polyphyletic origin of eugregarines from different archigregarine lineages (Cavalier-Smith, 2014). To eliminate this ambiguity, we propose the term “epicytic crests” instead of “folds” for eugregarines and the terms “longitudinal folds” or “bulges” for archigregarines. The term “crests” has been already used to describe the eugregarine epicyte (Pitelka, 1963, p. 90) and corresponds well to their narrow shape, compressed from the sides, in contrast to the large, gently sloped pellicular folds of archigregarines.

Figure 11 Comparison of archigregarine (A–B) and eugregarine (C–F) cell organization with their main diagnostic characteristics (candidate synapomorphies).

(A and C) Cross sections of the cortex of a typical representatives showing regularly arranged longitudinal subpellicular microtubules (smt) in archigregarine longitudinal folds vs. ripple dense structures (apical arcs (aa)) and 12-nm filaments (apical filaments (af)) closely adjacent to the inner membrane complex (imc) of the pellicle within the tops of eugregarine epicytic crests; typically, internal lamina (il) forms links in the bases of the epicytic crests; pm, plasma membrane. (B) Archigregarine trophozoite showing a mucron (mu) with an apical complex (conoid (co) and rhoptries (rh)) and mucronal food vacuole (mv) performing myzocytosis (the cell junction type between the host and parasite cells is septate junction); the cytoplasm is rich in microneme-like organelles (mo). (D) Formation of the epimerite (ep) in eugregarines: a protuberance of the gregarine cell emerging ahead of the degrading apical complex. (E) Epimerite (so-called “mucron”) of some aseptate gregarines Lecudina spp. without the apical complex and with a large flat frontal vacuole and microtubules in the base. (F) Epimerite of septate gregarines with the same structures and with mitochondria. In eugregarines, the cell junction between the host and parasite is formed by two closely adjacent plasma membranes and there is no myzocytosis (or perhaps only in the earliest developmental stages before the reduction of the apical complex).

The hypothesis of eugregarine polyphyly is inferred solely from ambiguous SSU rDNA-based molecular phylogenies (which are low resolved and apparently affected by LBA) and assumes the independent origin both of epimerite and epicytic crests in the major eugregarine lineages, i.e., they are convergences (homoplasies), that appears unlikely considering their detailed ultrastructural resemblance in a broad range of gregarines (see below). Therefore, following the principle of Ockham’s razor (minimum of assumptions), we rather consider the epimerite and epicytic crests shared-derived characteristics of eugregarines (Figs. 10 and 11). Apart from the Ancoroidea (Ancora sagittata), these features are widespread within all other eugregarine superfamilies revealed to date: Actinocephaloidea (Baudoin, 1969; Vávra, 1969; Ormierès & Daumal, 1970; Vorobyeva & Dyakin, 2011), Stylocephaloidea (Desportes, 1969), Gregarinoidea (Devauchelle, 1968; Tronchin & Schrével, 1977; Dallai & Talluri, 1983; Schrével et al., 1983), Cephaloidophoroidea (epimerite is understudied) (Desportes, Vivarès & Théodoridès, 1977; Simdyanov, Diakin & Aleoshin, 2015), and Lecudinoidea (Schrével & Vivier, 1966; Vivier, 1968; Ouassi & Porchet-Henneré, 1978; Corbel, Desportes & Théodoridès, 1979; Simdyanov, 1995b, 2004, 2009; Diakin et al., 2016). It should be noted, however, that the Lecudinoidea and Actinocephaloidea, apart from typical (core) representatives, also include morphologically divergent forms possessing various modifications of the epicyte structure (fusion or reduction of the epicytic crests, sometimes formation of hair-like projections). These modifications are always attended by the loss of gliding motility and transition to metaboly (peristaltic motility) or nonmotility. Within the Lecudinoidea, such is the divergent family Urosporidae parasitizing coelom of polychaetes and echinoderms, unlike the Lecudinidae—core representatives, which are intestinal parasites, chiefly in polychaetes. Within the Actinocephaloidea, there are intracellular neogregarines without epicyte (Žižka, 1978) and the family Monocystidae parasitizing seminal vesicles of oligochaetes, whereas core representatives, the Actinocephalidae, are intestinal parasites of insects (chiefly). The monocystids show very divergent and various structure of the cortex and possess peristaltic motility (Miles, 1968; Warner, 1968; Vinckier, 1969; MacMillan, 1973) that is similar with some species of the Urosporidae (Dyakin & Simdyanov, 2005; Landers & Leander, 2005; Leander et al., 2006; Diakin et al., 2016). However, in terms of comparative anatomy, existence of certain divergent (aberrant) forms does not cancel the presence of a shared bauplan in core representatives of the group: e.g., “archiannelids” and leeches within annelids when compared with the core forms as polychaetes and oligochaetes—in this and many other cases, large majority of diagnostic characteristics may be applied only to the “core group” (e.g., although the annelids are monophyletic, it is impossible to indicate synapomorphies shared by all without exception representatives of the group). The core (non-aberrant) representatives of all known eugregarine lineages/superfamilies share both epimerite (understudied in Cephaloidophoroidea) and epicytic crests. Therefore, in compliance with the main principle of cladistics, we present a morphology-driven hypothesis on the monophyly of eugregarines based on the presence of the epimerite and epicytic crests as defining synapomorphies of the order Eugregarinida (Figs. 10 and 11), which may be included in its diagnosis (see below in the taxonomical subsection). This hypothesis cannot be tested by the currently available molecular data but is potentially consistent with it (at least, does not contradict: see Figs. 7 and 9). More robust molecular datasets are therefore needed to test whether these structures represents true homologies.

Three cases seemingly challenge the monophyly eugregarines at the morphological level: Veloxidium leptosynaptae, Caliculium glossobalani, and Seledinium melongena (Wakeman & Leander, 2012; Wakeman, Heintzelman & Leander, 2014; Wakeman et al., 2014). V. leptosynaptae and C. glossobalani broadly resemble archigregarines but SSU rDNA phylogenies unambiguously place them in the eugregarine clades Lecudinoidea and Gregarinoidea, respectively (Fig. 7). External morphology and ultrastructure and of S. melongena is somewhat similar to C. glossobalani, but it is a sister taxon to the archigregarine Selenidium terebellae (Fig. 7). The certain morphological resemblances of all three species have been used to challenge the archi- and eugregarine concepts, however, their ultrastructure provides no firm support for such conclusions. V. leptosynaptae lacks ultrastructural data altogether and those available for C. glossobalani do not reveal any key ultrastructural features of either archi- or eugregarines (see Figs. 10 and 11). C. glossobalani lacks a genuine mucron, the associated conoid, mucronal vacuole, and rhoptries and the layered arrangement of the subpellicular microtubules that is characteristic for archigregarines (see above). C. glossobalani also lacks eugregarine epicytic crests (it has only low, wide, and mildly sloping longitudinal folds resembling those in archigregarines, however without microtubules) and the epimerite: its sucker-shaped attachment organelle is covered by a trimembrane pellicle in detached individuals (note, however, that no trophozoites attached to the host cells were examined by TEM). Thus, as yet, V. leptosynaptae and C. glossobalani rather appear to be morphologically divergent eugregarines when compared with the typical representatives of their phylogenetic lineages (e.g., Lecudinoidea and Gregarinoidea)—possibly because they both occur in unusual habitats or hosts (compare with Urosporidae and Monocystidae)—but their similarity with archigregarines is superficial and not supported by ultrastructural data. The situation with S. melongena is somewhat similar to C. glossobalani: most aforementioned key features of archi- and eugregarines were not identified (Wakeman, Heintzelman & Leander, 2014). The presence of structures resembling the mucronal vacuole and micronemes or small rhoptries combined with molecular phylogenetic data nevertheless suggests that S. melongena could be a divergent archigregarine, which has undergone a morphological transformation possibly due its unusual, coelomic localization within the host. Certain ultrastructural similarities between C. glossobalani and S. melongena are actually caused by the “shared” absence of the defining ultrastructural features of both archi- and eugregarines. Altogether, the external morphology of V. leptosynaptae, C. glossobalani, and S. melongena reaffirms that the external morphology of gregarine trophozoites and gamonts is a poor taxonomic marker susceptible to convergence. Because evidence of the key ultrastructural characteristics in all three species is presently lacking, they cannot be used in evaluating hypotheses on the evolutionary origin of archi- and eugregarines.

The dichotomy between aseptate and septate gregrannies is rejected by the SSU rDNA phylogenies: that is consistent with the hypothesis of Grassé, which considered some aseptate forms likely derived secondarily from septate gregarines (e.g., Paraschneideria with young septate trophozoites and aseptate gamonts and, most likely, Ascogregarina (former “mosquito Lankesteria”)); also there are intermediate forms between aseptate and septate gregarines, e.g., Ganymedes (Grassé, 1953; Schrével & Desportes, 2013b). Hence, the septum appears to be an evolutionarily unstable trait, therefore the separation of the order Eugregarinida into Aseptata and Septata, which is additionally not supported by available molecular data, should be abolished.

In contrast, the separation of eugregarines into several deep lineages (superfamilies) is well supported by the SSU rDNA phylogenies, although some families of gregarines are still missing in these analyses (e.g., Dactylophoridae and Hirmocystidae), and others are represented by a single species (e.g., Monocystidae) or are composed exclusively of environmental sequences (e.g., the cluster of “Ammonia-like” clones). Despite these limitations, designation of the well supported eugregarine clades with a superfamily rank (Actinocephaloidea, Stylocephaloidea, Gregarinoidea, Cephaloidophoroidea, and Lecudinoidea) appears to be natural and has been proposed repeatedly (Clopton, 2009; Rueckert et al., 2011; Simdyanov & Diakin, 2013; Cavalier-Smith, 2014).

The morphology and host spectra of eugregarine superfamilies (Table 3) does not correlate with the rates of evolution of their SSU rDNAs. The ancestral eugregarines were likely intestinal parasites of marine invertebrates (similarly to archigregarines and lower coccidians), whose morphology may have resembled aseptate lecudinids with weakly developed epimerites. However, the Lecudinoidea have highly divergent sequences, whereas some taxa with short branched sequences have a complex morphology (Actinocephalidae and Stylocephalidae). Consequently, the use of general morphology of trophozoites in defining taxonomic levels lower than the order should be implemented with caution, because these characteristics may be convergent (e.g., peristaltic motility and aberrant surface structures in eugregarines that occur in the host coelom—see above). The independent morphological and molecular evolutions in eugregarines can be also observed in Ancora sagittata and its sister group Polyplicarium, which have aseptate organization resembling lecudinids, but are not closely related to the Lecudinoidea (Fig. 7). Since they form the firmly supported separated molecular phylogenetic lineage, we formally delimit them as a new superfamily Ancoroidea in the framework of Linnaean taxonomy.

Table 3 Characteristics of the main phylogenetic lineages of eugregarines1.

Lineage and main representatives	Main characteristics	Hosts	
Actinocephaloidea (short branch)	Morphologically diverse group, but well-supported with SSU rDNA phylogenies; possible morphological synapomorphy: biconical or bipyramidal oocysts2; also frontal syzygy are characteristic for the majority of the representatives	Chiefly insects, but also earthworms	
Actinocephalidae3	Septate, typically with well-developed protruded epimerit often bearing hooks or other projections, or secondary aseptate (e.g., Ascogregarina, Paraschneideria); gliding motility and typical epicyte; epimerite discarding in mature gamonts; syzygy frontal; oocysts chiefly biconical or bipyramidal (sometimes spiny), sometimes crescent (e.g., Menospora)	Insects (intestine)	
Monocystidae	Aberrant aseptate gregarines without pronounced epimerite (no valid TEM data); peristaltic motility (metaboly), aberrant epicyte (variously modified up to full loss); syzygy frontal or lateral; oocysts biconical	Earthworms (seminal vesicles and coelom)	
“Neogregarines”	Aseptate forms (sometimes intracellular) without pronounced epimerite; gliding motility and typical epicyte are absent in studied representatives; syzygy frontal (including intracellular species (Žižka, 1978)); oocysts biconical or bipyramidal (sometimes spiny)	Insects (intestine, Malpighian tubules, and fat body—for intracellular species)	
Stylocephaloidea (short branch) Stylocephalidae	Septate gregarines likely related to Actinocephaloidea: trophozoite and syzygy morphology similar to the family Actinocephalidae, but epimerite is always elongate, without projections; oocysts purse-shaped	Insects (intestine)	
Gregarinoidea (long branch)	Chiefly septate (excepting Caliculium glossobalani). Possible synapomorphy: gametocysts with sporoducts (tubular projections for the releasing of oocysts); the other non-sequenced gregarines having them (e.g., Gigaductus having merogony like neogregarines (Ormierès, 1971)) are probably members of this lineage (Simdyanov, 2007; Schrével & Desportes, 2015)	Chiefly insects (intestine)	
Gregarinidae4	Septate with bulbous epimerite retracted or condensed in mature gamonts, gliding motility and typical epicyte; early syzygy of caudo-frontal type; gametocysts with sporoducts, oocysts barrel-like	Insects (intestine)	
Leidyanidae	Similar to Gregarinidae, but with late syzygy (just before gametocyst formation)	Insects (intestine)	
Caliculium glossobalani	Weird marine aseptate gregarine superficially resembling Selenidium, but possessing neither bending motility nor the key ultrastructural features of archigregarines. Molecular data place it within Gregarinoidea	Glossobalanus minutus (Hemichordata), intestine	
Cephaloidophoroidea (extremely long branch)	Septate and aseptate forms, intestinal parasites in crustaceans, robust clade in molecular phylogenetic trees with multiple distinct signatures in SSU rDNA sequences; no obvious morphological synapomorphies	Crustaceans (intestine)	
Cephaloidophoridae	Septate, with small epimerite (cephaloid) separated by septa persisting in mature gamonts, gliding motility and typical epicyte; syzygy caudo-frontal; oocysts ovoid or spherical with equatorial suture or crest	Crustaceans (intestine)	
Uradiophoridae	Septate, with small epimerite persisting in mature gamonts, gliding motility and typical epicyte; syzygy caudo-frontal; oocysts spherical with equatorial crest or radial projections	Crustaceans (intestine)	
Thiriotiidae	Aseptate, epimerite appears absent (no TEM data), gliding motility and typical epicyte; syzygy of unusual type (head-to-side); oocysts unknown	Crustaceans (intestine)	
Ganymedidae	Aseptate, epimerite appears absent (no TEM data), gliding motility and typical epicytic folds, syzygy caudo-frontal; oocysts unknown	Crustaceans (intestine)	
Lecudinoidea (long branch)	Chiefly aseptate forms without obvious morphological synapomorphies, but robust clade in molecular phylogenetic trees with nice multiple signatures in SSU rDNA sequences	Broad range of various aquatic (chiefly marine) invertebrates	
Lecudinidae	Chiefly aseptate (but Ferraria, Sycia, Ulivina, and some others) with weakly developed sucker-like epimerite thought to be condensed in mature gamonts; gliding motility and typical epicyte; syzygy mainly lateral or frontal; oocysts ovoid	Chiefly polychaetes and related groups, and also tunicates (intestine)	
Urosporidae5	Aseptate with weakly developed attachment apparatus (no TEM data); motility can be modified from gliding to peristaltic or loss of motility; epicyt from typical to aberrant; syzygy mainly lateral or frontal; oocysts heteropolar with funnel on the one pole and tail-like projection(s) on the other pole	Chiefly coelom of echinoderms and polychaetes	
Ancoroidea (moderately long branch)	Robust clade in molecular phylogenetic trees (SSU rDNA); the external morphology of known representatives is similar to Lecudinidae; ultrastructure is understudied	Capitellid polychaetes (intestine)	
Ancoridae	Aseptate with two lateral projections and bulbous epimerite thought to be discarded in mature gamonts; gliding motility and typical epicyte, but apical filaments are probably modified; syzygy unknown; oocysts ovoid	Capitellid polychaetes (intestine)	
Polyplicariidae	Aseptate; attachment apparatus unknown; gliding motility and epicyte crests (no TEM data)	Capitellid polychaetes (intestine)	
“Ammonia-like” environmental SSU rDNA sequences (moderate length branch)	Identified only with molecular data (SSU rDNA). Putative gregarines, expected to be aseptate, possibly a part of the current Lecudinidae	Unknown	
Notes:

1 Morphological characteristics were taken mainly from Grassé (1953) and Perkins et al. (2000).

2 Oocysts = sporocysts or spores in (Grassé, 1953; Schrével & Desportes, 2013a; Schrével et al., 2013).

3 Sensu lato, i.e., including Sphaerocystidae and other related minor families separated by Levine (1985, 1988).

4 Sensu lato, including Blabericolidae (Clopton, 2009).

5 Sensu lato, including Gonosporidae (Schrével & Desportes, 2013b).

Molecular and morphological diversity in Ancoroidea

All environmental sequences in GenBank, which were affiliated with the Ancoroidea (Fig. 8), were obtained from anoxic marine sediments, including cold methane seeps and shallow water hydrothermal zones (Edgcomb et al., 2002; Stoeck & Epstein, 2003; Stoeck, Taylor & Epstein, 2003; Stoeck et al., 2007; Takishita et al., 2007; Santos et al., 2010; Boere et al., 2011; Garman et al., 2011; Orsi et al., 2012). The geographical distribution of the samples containing the ancoroid sequences is wide: arctic, temperate, and tropical zones of the Atlantic and Indo-Pacific regions: Greenland, North America (Vancouver (BC) and Cape Cod), the Gulf of Mexico, and Papua New Guinea; however, the sequences that are closely related to Ancora sagittata were collected only from the Atlantic and European Arctic. Considering that both Ancora sagittata and Polyplicarium spp. parasitize polychaetes in the family Capitellidae and that most of related environmental sequences have been retrieved from anoxic environments, in which the Capitellidae are preferentially distributed, we hypothesize that all these Ancoroidea likely share the same group of hosts in similar habitats.

In this context, the affiliation of another species, Ancora prolifera Clausen, 1993, to this genus is questionable because this species is a parasite of the non-capitellid polychaete Microphthalmus ephippiophorus (Hesionidae). Ancora prolifera and Ancora sagittata are morphologically similar: the latter also has lateral projections; however, these are not located in the plane of the body axis but at an angle to it, similar to lifted wings of a bird (Clausen, 1993). Clausen also observed a nucleus-like structure in these projections (apart from the genuine nucleus) and therefore proposed that cell division in this gregarine occurs via budding, which has never been observed in eugregarines. One additional species, Ancora lutzi Hasselmann, 1918, was only described in a preliminary note (Hasselmann, 1918) without figures and delimitation of type material. The gregarines were present in two individuals of Capitella capitata (the same host species as Ancora sagittata) collected in the bay of Manguinhos (Brazil) and distinguished from Ancora sagittata by a shorter and wider body, more intense granulation in the cytoplasm, and a frontal nucleus. This species was never rediscovered and was later suggested to represent a morphological variant of Ancora sagittata (Watson Kamm, 1922).

Because the Ancoroidea is split into two distinct clusters (Fig. 8), we recognize two families within this superfamily: Ancoridae fam. nov. and Polyplicariidae Cavalier-Smith, 2014. The family Ancoridae is currently monotypic (single genus Ancora). This taxonomical rearrangement removes Ancora sagittata from the family Lecudinidae. The family Polyplicariidae, apart from the type genus Polyplicarium, likely includes at least two additional undescribed genera corresponding to two environmental clusters (Fig. 8). The representatives of Ancoroidea display small morphological differences from the Lecudinidae in the fine structure of the epicytic folds and attachment apparatus (in Ancora sagittata described above; the ultrastructure of Polyplicarium is not known).

Putative cryptic species in Ancora sagittata

Considerable differences have been observed between two ribotypes of Ancora sagittata, WSBS 2010 contig (ribotype 2) and WSBS 2006, 2011, and Roscoff contigs (ribotype 1). Four CBCs in ITS2 (Fig. 6) suggest that these ribotypes represent two distinct cryptic species (Coleman, 2000, 2009; Müller et al., 2007; Wolf et al., 2013). Although the ribotype of the Ancora sagittata type material is not known, we still have annotated the sequences of ribotype 1 as belonging to the type species Ancora sagittata (GenBank accessions KX982501–KX982503) because they appear more widespread then ribotype 2, the only sequence from the sample WSBS 2010, which was annotated as Ancora cf. sagittata, KX982504.

Nine environmental sequences closely related to Ancora sagittata may belong to other cryptic species within the same morphotype since, in the tree, they are flanked by the sequences obtained from the same morphospecies, although belonging to the different ribotypes (Fig. 8). Two other environmental sequences, M60E1D07 and M23E1H07, which form a sister branch to the Ancora sagittata cluster, putatively belong to another species in the genus Ancora.

Hasselmann (1927) proposed parthenogenetic formation of oocysts (solitary encystment of gamonts) in Ancora sagittata based on the behavior of solitary mature gamonts due to similarities to late syzygy in other gregarines (flexion of the body and circular gliding (rotation)). Although solitary encystment and gametogenesis have not been described in Ancora sagittata, it is possible that some of its morphospecies exist as parthenogenetic clones, while others likely have a regular sexual cycle. This possibility could explain branch length differences within the Ancora sagittata group (Fig. 8) and the sympatric coexistence of two cryptic species (ribotypes) of Ancora sagittata at the WSBS. Thus, the hypothesis of a cryptic species complex in Ancora sagittata should be reconsidered both in terms of reproduction modes and the presence of a species complex in its host, Capitella capitata (Grassle & Grassle, 1976).

Taxonomic actions: modification of gregarine and eugregarine diagnoses and establishment of the new superfamily Ancoroidea

Phylum Apicomplexa Levine, 1970

Subphylum Sporozoa Leuckart, 1879

Class Gregarinomorpha Grassé, 1953, emend.

Diagnosis. Sporozoa. Gamont coupling (syzygy) followed by encystment (formation of gametocyst); progamic mitoses in both gamonts; gametogenesis and fecundation within the gametocyst; anisogamy is characteristic: female gametes are non-flagellated, male gametes usually flagellated, bear 1 flagellum; oocysts without sporocysts (sporozoites lie free within the oocyst, not in its internal compartments). Typical representatives are epicellular intestinal parasites of invertebrates, mainly Trochozoa, Arthropoda, and Deuterostomia, including lower Chordata (Tunicata).

Order Eugregarinida Léger, 1900, emend.

Diagnosis. Gregarinomorpha. Typically: gliding locomotion of the gamonts likely provided by epicytic crests, i.e., longitudinal pellicular folds of complex structure (rippled dense structures (apical arcs) and 12 nm apical filaments within the tops of the crests, the links of internal lamina in their bases); the attachment apparatus is chiefly an epimerite that develops ahead of the sporozoite apical complex, which disappears in the beginning of trophozoite formation; the epimerite is mostly absent in mature gamont (degenerated, retracted, or discarded). A number of representatives exhibit a septate morphology of the trophozoites: there are one or more fibrillar septa that separate the cell into compartments—protomerite and deutomerite.

Note 1. The morphological synapomorphies of the Eugregarinida compared with the plesiomorphies of Archigregarinida have been presented as diagrams in Fig. 11.

Note 2. There are a number of aberrant representatives, which lose the typical structure of the attachment apparatus and epicyte. This is frequently correlated with the transition from the intestinal to coelomic parasitism (e.g., Monocystidae and Urosporidae).

Note 3. The order includes several superfamilies (see below), which were erected after molecular phylogenetic analyses of SSU rDNA. However, recent molecular data do not encompass the complete taxonomical diversity of eugregarines, and we expect additional superfamilies to be established in the future. The current composition and characteristics of the superfamilies described to date are consistent with the characteristics of corresponding molecular phylogenetic lineages (Table 3).

Superfamily Actinocephaloidea

Note. Since molecular data corroborate the assumption of Grassé about the origin of neogregarines from actinocephalids, the superfamily must also include neogregarines. Consequently, the order Neogregarinida should be abolished.

Superfamily Stylocephaloidea

Superfamily Gregarinoidea

Superfamily Cephaloidophoroidea

Note. This name was first proposed in Rueckert et al. (2011), but it was changed by Cavalier-Smith (2014) into Porosporoidea because of the earlier establishment of the family Porosporidae Labbé, 1899 than Cephaloidophoridae Kamm, 1922. However, considering that the guidelines of the International Code of zoological nomenclature have a recommendatory (suggestive) nature for superfamilies, the name Cephaloidophoroidea can be accepted and appears to be more appropriate: the SSU rDNA of Cephaloidophora communis, the type species of this family, was sequenced (Rueckert et al., 2011), unlike type species of all of the other families included in this clade. Additionally, these families are more or less problematic and require revision: e.g., at the last time, the family Thiriotiidae was separated from the Porosporidae based on the shape of the unusual syzygy (Schrével & Desportes, 2013b); the DNA sequences of the true representatives of the Porosporidae (Porospora, Nematopsis) are unavailable.

Superfamily Lecudinoidea

Note. Cavalier-Smith (2014) used the name Urosporoidea, but Lecudinoidea appears to be more appropriate because the SSU rDNA sequence of the type species of the family Lecudinidae, L. pellucida, is available. In contrast, DNA sequences of the type species of the family Urosporidae, Urospora nemertis, are unavailable. Additionally, the Urosporidae is the aberrant family (see above), which also could present nomenclatural problems: while other urosporids chiefly parasitize the coelom of echinoderms and polychaetes, the type species is an intestinal parasite of the nemertean Baseodiscus delineatus, and its taxonomical position and status may be questionable.

Superfamily Ancoroidea, superfam. nov.

Diagnosis. Eugregarinida. Aseptate forms parasitize marine polychaetes, mainly the family Capitellidae; tightly adjacent epicytic crests; gliding motility. Molecular data: the robust SSU rDNA clade.

Note. For more grounded diagnoses of the entire group and subgroups within it, additional data are necessary, e.g., the ultrastructure of Polyplicarium spp.

Family Polyplicariidae Cavalier-Smith, 2014

Diagnosis (preliminary). Ancoroidea. Characteristics of the type genus Polyplicarium.

Genus Polyplicarium Wakeman et Leander 2013. Ovoid to elongate trophozoites with a blunt anterior end. The posterior end is either blunt or tapers to a point. Longitudinal epicytic folds with a density of 4–5 per 1 μm; most trophozoites also have a distinct region of wider, shallower epicytic folds; gliding locomotion; other life-cycle stages are unknown. There are four named species.

Note. The family likely includes at least two additional undescribed genera that are represented only by environmental sequences.

Family Ancoridae Simdyanov, fam. nov.

Diagnosis. Ancoroidea. Monotypic, characters of the type genus Ancora.

Genus Ancora Labbé, 1899. Trophozoites and gamonts with two lateral projections giving them appearance of an anchor. Gliding locomotion. Growing trophozoites with a bulbous epimerite. Syzygy unknown. Simple gametocyst dehiscence by rupture. Oocysts ovoid. There are three named species, but two of them are questionable.

Note. The type morphospecies Ancora sagittata (Leuckart, 1860) Labbé, 1899 likely is a complex of cryptic sibling species.

Conclusion

The results of our work point to several new directions of importance to gregarine research. The molecular phylogenies based on the SSU rDNA alone firmly delimit several major lineages (superfamilies) in eugregarines but not their suborders (Aseptata and Septata), a finding that is more consistent with Grassé’s taxonomical scheme (Grassé, 1953) than with the current taxonomy established by Levine and the followers (Levine, 1985, 1988; Perkins et al., 2000). The results also corroborate other Grassé’s assumptions (Grassé, 1953): (i) the polyphyletic origin of neogregarines, likely from different representatives of the eugregarine family Actinocephalidae; (ii) the secondary origin of some aseptate gregarines from septate ancestors; and (iii) the importance of gregarine co-evolution with their hosts. The molecular evidence indicates that both the life cycle peculiarities (presence or absence of merogony) and the general morphology of eugregarine trophozoites (septate or aseptate), which are broadly employed in the current eugregarine taxonomy, are unreliable. However, SSU rDNA phylogenies do not resolve their deeper branching and do not allow for testing the monophyly of Eugregarinida, Archigregarinida, and all gregarines, possibly due to their explosive evolutionary radiation and/or rapid sequence evolution that resulted in numerous long branches in molecular phylogenies suffering from long-branch attraction (LBA) artefacts.

The near-complete rDNA operon likely provides an increased resolution over SSU rDNA and appears more resilient to LBA (Simdyanov, Diakin & Aleoshin, 2015; Fig. 8). Although neither of the markers resolves deep relationships among gregarines recently, a representatively sampled rDNA operon is likely to provide a more reliable test of the group’s morphological evolution in the future. The best strategy for the development of gregarine phylogeny (sensu lato) and high-rank taxonomy seems to be reconciling and combining morphological evidence with unambiguous molecular data such as well-resolved deep branching in the molecular phylogenetic trees of gregarines—probably with the use of concatenated nuclear markers compiled from transcriptomic and genomic datasets. However, until such datasets become available, we propose to treat the shared ultrastructural characteristics of their epicytic crests and epimerite as synapomorphies of eugregarines and, consequently, as evidence for their monophyly by following the principles of cladistics. At the same time, relying on the firm molecular phylogenetic support and following previous works, we also propose to abolish the suborders Aseptata and Septata within the order Eugregarinida (Simdyanov & Diakin, 2013; Cavalier-Smith, 2014) and accept the robust molecular phylogenetic lineages as superfamilies instead (Clopton, 2009; Rueckert et al., 2011; Simdyanov & Diakin, 2013; Cavalier-Smith, 2014). On the same ground, we acknowledge the abolition of the order Neogregarinida (Simdyanov & Diakin, 2013; Cavalier-Smith, 2014), which apparently comprises divergent representatives of the eugregarine superfamily Actinocephaloidea. Finally, following this assertion, we also propose to remove the absence of merogony from the diagnostic criteria of eugregarines, despite that the current gregarine taxonomy relies heavily on this characteristic (Levine, 1985; Perkins et al., 2000; Adl et al., 2012). The majority of these proposals receive molecular phylogenetic and ultrastructural backing and although some are more preliminary than others (the monophyly of eugregarines will require thorough testing, e.g., by evidence from multigene molecular phylogenetic analyses), they altogether represent a next step in a much needed revision of the gregarine taxonomy and evolution.

Supplemental Information

Supplemental Information 1 Ancora.

Raw data: DNA sequences.

Click here for additional data file.

Supplemental Information 2 Raw data on SSUrDNA phylogenies.

Raw data. The ZIP file contains two compressed folders (MrBayes and RAxML) comprising input, log, and output files as well as resulting phylogenies of two phylogenetic programs: MrBayes and RAxML, respectively. For MrBayes, its input file (infile.nex) contains matrix (cut down nucleotide alignment) and parameters of computations, its log files (*.log) contains results of computations and summarizing of parameters, its tree files (*.tre) contain resulting phylogenies. The folder “MrBayes” also includes one subfolder “Bootstraps” containing bootstrap supports for the resulting Bayesian tree using bootstraps calculated by RAxML program with the same alignment. For RAxML, there are three subfolders (“Bootstrap”, “ML”, and “Tree”, containing results of bootstrap calculations, resulting tree of ML analysis, and the resulting tree with bootstrap supports, respectively. Also there are input files (“infiles”) in these folders subfolders containing the same alignment “cut down for programs”, *.info files containing parameters, and summarized results of the computations.

Click here for additional data file.

Supplemental Information 3 Raw data on LSU rDNA phylogenies.

Raw data. The ZIP file contains two compressed folders (MrBayes and RAxML) comprising input, log, and output files as well as resulting phylogenies of two phylogenetic programs: Mr Bayes and RAxML, respectively. For MrBayes, its input file (infile.*) contains nucleotide alignment and parameters of computations, its log files (*.log) contains results of computations and summarizing of parameters, its tree files (*.tre) contain resulting phylogenies. The folder “MrBayes” also includes one subfolder “Bootstraps” containing bootstrap supports for the resulting Bayesian tree using bootstraps calculated by RAxML program with the same alignment. For RAxML, there are three subfolders (“Bootstrap”, “ML”, and “Tree”, containing results of bootstrap calculations, resulting tree of ML analysis, and the resulting tree with bootstrap supports, respectively. Also there are infiles in these subfolders containing the same alignment, *.info files containing parameters and summarized results of the computations.

Click here for additional data file.

Supplemental Information 4 Raw data on ribosomal operon (concatenated SSU, 5.8S, and LSU rDNA alignments) phylogenies.

Raw data. The ZIP file contains two compressed folders (MrBayes and RAxML) comprising input, log, and output files as well as resulting phylogenies of two phylogenetic programs: Mr Bayes and RAxML, respectively. For MrBayes, its input file (infile.*) contains nucleotide alignment and parameters of computations, its log files (*.log) contains results of computations and summarizing of parameters, its tree files (*.tre) contain resulting phylogenies. The folder “MrBayes” also includes one subfolder “Bootstraps” containing bootstrap supports for the resulting Bayesian tree using bootstraps calculated by RAxML program with the same alignment. For RAxML, there are three subfolders (“Bootstrap”, “ML”, and “Tree”, containing results of bootstrap calculations, resulting tree of ML analysis, and the resulting tree with bootstrap supports, respectively. Also there are infiles in these subfolders containing the same alignment, *.info files containing parameters and summarized results of the computations.

Click here for additional data file.

Supplemental Information 5 SSU alignment.

The initial alignment of SSU (18S) rDNA containing alignment mask. Can be read with BioEdit program.

Click here for additional data file.

Supplemental Information 6 SSU rDNA alignment cut down for programs.

The cut down (minimized to mask) alignment (matrix), which was further used to create the input files for computer programs. Can be read with BioEdit program.

Click here for additional data file.

Supplemental Information 7 5.8S rDNA Ancora.

Initial alignment of 5.8S rDNA containing alignment mask.

Click here for additional data file.

Supplemental Information 8 LSU Ancora.

Initial alignment of LSU (28S) rDNA containing alignment mask.

Click here for additional data file.

Supplemental Information 9 LSU rDNA alignment cut down for programs.

The cut down (minimized to mask) alignment (matrix) of LSU rDNA, which was further used to create the input files for computer programs. Can be read with BioEdit program.

Click here for additional data file.

Supplemental Information 10 The concatenated ribosomal operon alignment cut down for programs.

The cut down (minimized to mask) alignment (matrix), which was further used to create the input files for computer programs. Can be read with BioEdit program.

Click here for additional data file.

Supplemental Information 11 Table S1. Results of pairwise comparison of Ancora sagittata sequences with each other and with Polyplicarium spp.

Direct pairwise comparison of Ancora sagittata sequences with each other (separately, SSU rDNA alone and concatenated SSU, 5.8S, and LSU rDNAs) and of A. sagittata and Polyplicarium spp. SSU rDNA sequences. The lengths of the overlaps in the alignment, mismatches (substitutions and indels), and percentage of identity of overlapping regions are shown (1st, 2nd, and 3rd lines of the cells, respectively). The diagonal of the table indicates the sequence total lengths. For A. sagittata sequences, two values are given: for SSU rDNA alone and for complete contigs without external spacers (ETS).

Click here for additional data file.

Supplemental Information 12 Table S2. Results of direct pairwise comparison of Ancoroidea sequences with each other.

The lengths of the overlaps in the alignment, mismatches (substitutions and indels), and percentage of identity of overlapping regions are shown (1st, 2nd, and 3rd number in cell, respectively).

Click here for additional data file.

This study utilized the CYPRES Science Gateway (Miller, Pfeiffer & Schwartz, 2010; http://www.phylo.org) and the Chebyshov Supercomputer Center of Lomonosov Moscow State University (http://parallel.ru/cluster) to perform the phylogenetic computations. DNA sequencing (Sanger) was performed at the DNA sequencing center “Genome” (Engelhardt Institute of Molecular Biology, Russian Academy of Sciences, www.genome-centre.ru). The electron microscopy studies were performed at the Laboratory of electron microscopy of Faculty of Biology, Lomonosov Moscow State University, and at the Centre of Electron Miscroscopy of I.D. Papanin Institute for Biology of Inland Waters, Russian Academy of Sciences. The authors are grateful to Dr. Gulnara Mirzayeva (Institute of Gene Pool of Plants and Animals, Uzbek Academy of Sciences, Republic of Uzbekistan) for her assistance carrying out the experimental work. We thank Maria Logacheva (Belozersky Institute of Physico-Chemical Biology, Russia) for the help with genomic sequencing. Many thanks are extended to Prof. Isabelle Florent and Dr. Isabelle Desportes (Muséum National d’Histoire Naturelle, France) for their valuable comments. We thank to Kevin Wakeman (Hokkaido University, Sapporo, Japan) and Jan Janouškovec (University College London, UK) for proofreading and commenting on the manuscript.

Additional Information and Declarations

Competing Interests

Author Contributions

DNA Deposition

Data Availability

The authors declare that they have no competing interests.

Timur G. Simdyanov conceived and designed the experiments, performed the experiments, analyzed the data, wrote the paper, prepared figures and/or tables.

Laure Guillou conceived and designed the experiments, performed the experiments, contributed reagents/materials/analysis tools, wrote the paper, reviewed drafts of the paper.

Andrei Y. Diakin performed the experiments, reviewed drafts of the paper.

Kirill V. Mikhailov conceived and designed the experiments, performed the experiments, analyzed the data, reviewed drafts of the paper.

Joseph Schrével analyzed the data, wrote the paper, reviewed drafts of the paper, collection of material (Ancora sagittata) in Roscoff (France), expert comments to gregarine morphology and taxonomy.

Vladimir V. Aleoshin conceived and designed the experiments, analyzed the data, contributed reagents/materials/analysis tools, reviewed drafts of the paper.

The following information was supplied regarding the deposition of DNA sequences:

The newly obtained rDNA sequences are accessible via GenBank accession numbers KX982500 to KX982504.

The following information was supplied regarding data availability:

The raw data (alignments, input, and output files of phylogenetic programs containing nucleotide alignments, parameters, and summarized results of computations) are included as Supplemental Dataset Files.

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
