# Peer review of "A new view on the morphology and phylogeny of eugregarines suggested by the evidence from the gregarine Ancora sagittata (Leuckart, 1860) Labbé, 1899 (Apicomplexa: Eugregarinida)"

_PeerJ, doi:10.7717/peerj.3354_

## Round 0.1 · original submission · Major Revisions

· Academic Editor

Major Revisions

I have heard from two reviewers regarding your manuscript. While their conclusions are widely different (minor revisions and reject), when you examine the comments you can see both reviewers do have some common ground. Reviewer 1 would like extensive editing, reduction of text, and more information (e.g. GenBank numbers) to make the study reproducible, with also more care in considering large conclusions, particularly of a taxonomic nature, while reviewer 2 suggests many English edits (reviewer 1 also mentions this), and also wants more information.

Please ensure your resubmitted text is edited by a native English-speaking scientist familiar with apicomplexans and phylogenies.

Overall, both reviews seem fair and very constructive to me, and they entail a considerable amount of work on your part to meet their comments and questions, but doing so will undoubtedly improve your submission. Therefore, my decision is "major revisions" are needed.

Reviewer 1 ·

Basic reporting

no comments (see pdf)

Experimental design

no comments (see pdf)

Validity of the findings

no comments (see pdf)

Additional comments

no comments (see pdf)

Annotated reviews are not available for download in order to protect the identity of reviewers who chose to remain anonymous.

·

Basic reporting

Basic Reporting:
As far as I can tell, the paper adheres to all PeerJ policies. Overall the paper was well written, but here a few typos and suggestions that should be addressed.

Abstract:
line 31 “in order to outline some” please change to “to outline new approaches”
line 40-1 “into which A. sagittata was previously belonged” should read “in which A. sagittata previously belonged ”
Introduction:
Line 62: “Protistean” I admire the authors for the ambitious task of modifying clades within the gregarines by discarding non-monphyletic clades in favor of monophyletic clades. However, Protistean is far from monophyletic, but perhaps used here as it is so firmly entrenched in people’s minds? It is important to mention Apicomplexa are unicellular eukaryotes within Alveolata, but I am not sure protist needs to be stated.

Line 64: “chiefly representatives of early branching lineages like gregarines ” is a bit awkward. Please change to “such as gregarines”
line 65: “are obligatory parasites ” should be “obligate”
line 69: “large unicellular organisms that ” please change to “large and range”
line 88 “Sporogony occurs in the environment, in which the ”. Do you mean after? If so, please change “in which” to “after”.
line 109: “lower representatives of the order”. I am not sure what this means?
Line 119: “core eugregarines ” What are core eugregarines? excluding blastogregarines? If not used in a phylogenetic sense, perhaps “most” would work?
Line 181: “Large ”. Missing an “A” before large.
Line 213: “This work also allowed to perform a more general overview of eugregarine phylogeny” should read. “This work allowed a more general”.
Methods: Lines 254-255: I uncertain, but there were 4 pooled molecular samples taken and used in all downstream phylogenetic analyses?
Line 257: “The nucleotide sequences ” please specify.
Line 258-9: please state why WGA was used here and only in some of the samples.
Line 261: “in several PCRs ” This is a bit too vague. I am having a hard time deciphering this section. Why were different primer sets used for different samples? was it to prevent amplification of the hyperparasites (as mentioned below)?
Lines 274-284: Please transfer to a table given the number of primers used. Same with citations provided in lines: 289-292

Line 294: I am not certain why some samples were treated differently here. Fragments II and IV were both cloned and not cloned prior to sequencing? What was the point of this?

Lines 297-8: not needed.

Lines 303-4: “This procedure has been performed using the same“. Does this mean the process for determining the LSU of the ciliate? Or this procedure has previously been published.
Line 318: why was a different DNA extraction method used for Lecudina?
Line 355: You state there are 4 alignments, but SSU, LSU and operon are only 3.
Line 358: “performing the computer calculations” please change to phylogenetic analyses.
Line 381: “identify environmental sequences” that are related to Ancora sp.?
Line 384: “environmental sequences were selected from the NCBI GenBank database using NCBI BLAST” Not needed.
Results: line 437: please remove the word “them”
Lines 502-3: “but with low support (PP=0.37, BP=12%)” This is no support in my opinion. I would not mention this clade at all.
Line 504: “erected at the superfamily level” Is this from Cavalier Smith 2014? Please include citation
Line 541: “belonged to this clade. ” Putative clade at best. With 47% pp and 17 bp support, I would argue there is no evidence for this being a clade in this dataset.
Line 556: perkinsisds “ is misspelled.
Discussion: Line 631: “nipping the host cell” I am not sure I follow what this means. Could you try another verb?
Line 635-6: “therefore an epimerite and not a mucron, although this is an aseptate gregarine ” this is a bit out of my specialty, but my understanding from this manuscript and my knowledge of the literature this is a novel hypothesis being advanced here. Please indicate here that this is such (i.e. a novel hypothesis) and a recommendation to differentiate the archi- from Eugregarines.
Line 671: ““pseudomucrons”.” I am confused by the use of this term. Here you argue that Eugregarines share a synapomorphy of a Epimerite and Archigregarines share a synapomorphy of a mucron. Therefore, it seems more logical to name (or rename) this structure a “pseudoepimerite”. This would be more consistent with the evolutionary hypotheses advanced here.
Line 678: “the other gregarines form a monophyletic lineage, although it was weakly supported” I agree for the SSU phylogeny, but not the LSU. I would add that this is just for this marker, because I agree with your argument in lines 688-9.
Line 698-700: “On the contrary, eugregarines undoubtedly appear to be a monophyletic group because core representatives of all their main lineages demonstrate at least two shared-derived characteristics, i.e., synapomorphies:” I would change to “likely” as the SSU shows very little support for this hypothesis. The LSU tree does not actually include the archigregarines so it is not possible to confirm this hypothesis.
Line 736: “considering” I would change to “despite” as it more accurately fits your description in my opinion.
Lines 747-8: “apparently resembled lecudinids” I would argue that it “may have” resembled this. The molecular evidence is equivocal.
Lines 754-5: “they exhibit a simple trophozoite morphology (e.g., Lecudinidae) that can be considered morphostasis” Again, this could be secondarily simplified. The molecular data are not convincing. This can be advanced as a hypothesis to be further examined.
Line 803: “did not designed as separate clusters ” Please change to “did not monophyletically cluster”
Line 839: Behavior is misspelled.
Figure 6: Rybotype should be Ribotype.

Experimental design

In this paper, the authors take on an ambitious task of refining gregarine taxonomy based on molecular phylogenetics and EM ultrastructure. This includes demonstrating that the major clades that comprise Eugregarinorida (itself the most speciose clade), are non-monophyletic and attempts to provide monophyletic groups associated with taxonomic descriptions. Given the lack of attention given to Gregarines compared to other members of Apicomplexa, this paper provides additional hypotheses regarding the evolution of this group. There are a couple of places in the methods that I had trouble understanding. Occasionally this was cleared up later in the results or discussion. Please find below specific examples of where I had trouble understanding what was done in addition to comments regarding the design and results themselves.

Line 261: “in several PCRs ”. I am not entirely certain why this was done? My understanding is that this was to create the entire rDNA operon by amplifying it piecemeal. However, this was only done for one species in the phylogeny of LSU and the entire operon.
Line 342: “Predicted secondary structures of ITS2” Why was this analysis done? and what were the parameters used in mfold? It should be explained here. From the figures later it appears it was to determine if the ITS2 was a valid structure with CBC and not a pseudogene. Please explain why here.
Line 347: “calculations of pairwise matched unambiguous nucleotide sites and indels in overlapping regions was performed” What was the point of this and what specifically was calculated? Please expand upon this. From the table it appears to be pairwise numbers of similar bases and mismatches.
Line 364-7: We first tried to avoid the inclusion of long branches in outgroups (excluding the gregarines, the majority of which are represented by long branches), and then we tried to provide an advantage to those species for which not only SSU, but also LSU and 5.8S rDNA, sequences are known“ So did you exclude all other taxa (outside the gregarines) with long branches? Which did you remove? I am not certain what you mean in the second part here. How can you provide an advantage to those with longer sequences? Where some of the sequences including truncated in the alignment. The figure legends of trees (fig. 6 and 7) state that they are SSU phylogenies? Does Figure 6 also contain LSU and 5.8S sequence? Or is these data only shown in Fig 9a and b?
Line 369-371: ”we used the LSU rDNA sequences of six unidentified species for the analysis, relying on the similarity of their SSU and LSU rDNA sequences to certain previously identified representatives of these clades“ I am concerned by this line. Does this mean that the alignment contains SSU and LSU genes aligned to each other or were they concatenated prior to alignment so that the SSU genes are aligned to each other and the LSU genes are aligned to each other? If the former explanation, I don’t feel comfortable with this analysis as you are comparing non-homologous genes in a phylogeny. If the latter, please further explain this. I imagine that there is little sequence data available in GenBank for the LSU of gregarines outside the ones generated here.
Figure 9: I am extremely excited by the high bootstrap support of the clades of apicomplexans in this tree. Although few LSU sequences exist for apicomplexans (especially within the gregarines), these data indicate the potential of this phylogenetic marker.
Line 600-1: “In general, the terminology used for the gregarine attachment apparatus is rather confusing.” I couldn't agree more! This paper is a great start to the conversation of finding true synapomorphies among the groups that mirror the molecular phylogenetics. One of the more exciting aspects of this manuscript is that these revisions to the ultrastructural features may make this a bit easier when describing the species both molecularly and morphologically.
Lines 713-5: “Therefore, to eliminate ambiguity, we recommend the term “epicytic crests” instead of “folds” for eugregarines and the terms “longitudinal folds” or “bulges” for archigregarines”. I like this idea in principle, but there is limited molecular support for the eugregarine/archigregarine split in this data set. I would advance these changes as hypotheses that should be tested with future phylogenetics (potentially involving LSU sequences from archigregarines).

Validity of the findings

I agree that the septate and aseptate Eugregarines should be dropped as taxonomic groups as they appear as non-monophyletic in several phylogenetic analyses. Likewise, neogregarines are also partially defined by host taxonomic range (i.e. only insect hosts). Several papers have now shown that they are not monophyletic. I also really like that the authors utilized a gene other than SSU to build their phylogenetic evidence as this is sorely lacking in this field. Especially given the high variability of SSU in the gregarines. I wish there had been archigregarines in the LSU tree as this would have provided more support for the Eugragarine/archigregarine split hypothesized in the morphological data, but I understand the effort required design primers to amplify these parasites to the exclusion of hosts and hyperparasites. Overall this is a very interesting paper that is supported by morphological ultrastructure and, to some degree, by the molecular data as well.

Additional comments

As stated above, I think this paper is an important contribution to the field and addresses a comparatively little studied clade within Apicomplexa. I recommend publication in PeerJ once these revisions are considered.

---

## Round 0.2 · Major Revisions

· Academic Editor

Major Revisions

Thank you for your patience with the reviewing process. In addition to the comments from the two reviewers as noted in the email below, I have received additional comments from reviewer #1 as follows (after they viewed your alignment files - thank you for sending them):

****Maybe it is just me, but I usually have a "working file" that I edit, and a "cut down program file" that gets run through the analysis program. And from these two files, people can see the addition-subtraction, and the exact file that got put through the program. To make this reproducible, those two files should be made available.

In the end the 18S and 28S alignments have a lot of gaps and taxa missing, and the Gregarine information is just all over the place, and one taxa is even missing almost 50% of the data in the 28S dataset. It's nice they got the data, and the data should be published because it can be made available that way. It just needs to come with a giant disclaimer of long branch attraction and taxon sampling and not a taxonomic revision. To not waste everyone's time, they can do this while addressing the review. I can almost guarantee I will get a tree similar to theirs (in that the values will be low). But what I will likely see with his datasets is that branching orders are going to be variable, and some of these taxa are going to start jumping around the tree (this happens to me all the time). And these datasets are just not something on which to rest a new view of evolution within this group.****

Thus, while reviewer #2 is now OK with the revised version, reviewer #1 still has some questions and issues that need to be addressed. In particular, they feel that the proposed taxonomic revision is preliminary and needs more disclaimers. In your responses, please let me know if you agree or disagree with this view, and support your statements with data or evidence. I look forward to seeing a revised version.

Reviewer 1 ·

Basic reporting

see pdf

Experimental design

see pdf

Validity of the findings

see pdf

Additional comments

see pdf

Annotated reviews are not available for download in order to protect the identity of reviewers who chose to remain anonymous.

·

Basic reporting

Basic Reporting:
This is the second time I have reviewed this paper and the authors have addressed all my concerns and further clarified sections that made the generation of the data more obvious. As far as I can tell they have done the same with the previous reviewer as well. I agree with Reviewer 1 that the monophyly of eugregarines needs to be based entirely on morphological data as I think we all believe the 18S data in Fig. 7 is equivocal (and LSU lacking). I agree that you have provided morphological data that suggests monophyly and hope you carry on with efforts to increase taxonomic breadth in the LSU phylogeny as I think it has strong potential for supporting or refuting this very hypothesis!

With that said, I do think that this paper should be published in PeerJ as it is an important step for simplifying morphological terminology and advancing phylogenetic hypotheses within the gregarines. There are a few additional typos listed below and a few opinions based on personal preference. These comments may be useful, or may not (again this is just my opinion and will NOT change the story or data). Also in my opinion, there is nothing that came up in the review process or in the revisions that change my view of the data or interpretations. Therefore, I, once again, recommend accepting this paper for publication in PeerJ.

As far as I can tell this paper still adheres to all PeerJ policies.

Abstract:
line 49-50 ", which is restricted to be used to archigregarines alone” is awkward. Maybe “which we recommend be restricted to archigregarines alone”
Introduction:
Line 145: “The taxonomy of gregarines remains largely incomplete”. I would argue that the “taxonomy and phylogeny” remains largely incomplete. Please consider adding.

Line 157-8: “The most productive taxonomical scheme of the gregarines is based on Grassé's hypothesis about their co-evolution with their hosts (Grassé, 1953). Following it, Grassé divided Schizogregarinida”
The problem with calling it the “most productive” taxonomy is that even Grasse recognized Schizogregarines as non-monophyletic. I don’t agree with the term “most productive” but would replace with “commonly used”, “formerly used” or something like that.

Line 159: “Achigregarinida” spelled wrong

Methods: Lines 1026-7: “ITS 2 is a peculiar genetic region” I appreciate the additional information regarding why this was done as it makes the data clearer. However, I am not sure that “peculiar” is the right word here. perhaps “fast-evolving” or something like that?

Lines 1264-1268: I would make clear here this was one of the four alignments mentioned above. The wording “we prepared another SSU rDNA” makes it seem like this is a fifth alignment.
Results: Lines 1344:“ which fitted” should be “which fit”
Lines 1538: “but with low support (PP=0.37, BP=12%)” This is no support in my opinion. I would not mention this clade at all.
ANSWER. Yes, this is no support values VALID for phylogenetic hypotheses (and we emphasized this each time in the Discussion; however in terms of statistics these are support values and they are low. What is wrong? We are saying the same thing I believe. In terms of the verbiage, to me “low support” is used to indicate low, but still valid, support. This may also have been a source of confusion for reviewer #1 as well? Perhaps a good compromise is “Eugregarines are separated from archigregarines and are monophyletic in both Bayesian and ML trees, but with little to no support (PP=0.37, BP=12%)”. I simply want to reiterate that these data are not convincing to the argument that there is monophyly of Eugregarines. I agree that the morphological data are convincing, but the limited taxonomic breadth of the LSU tree and the lack of support in the SSU tree cannot be used to validate this hypothesis one way or another as you seem to agree,
“Still, the SSU rDNA phylogeny is of little use for resolving the gregarine (and apicomplexan) branching order: all existing phylogenies show low support in the apicomplexan tree backbone. In particular, the monophyly of neither Eugregarinida nor Archigregarinida is not supported by published phylogenies, possibly due to an explosive evolutionary radiation of gregarines and/or rapid evolution of their SSU rDNA sequences.”
With that said, the hypothesis is still valid and supported by the morphological data. As mentioned above, I hope that you do carry through with the LSU data as this is quite promising as a marker… “I like this idea in principle, but there is limited molecular support for the eugregarine/archigregarine split in this data set. I would advance these changes as hypotheses that should be tested with future phylogenetics (potentially involving LSU sequences from archigregarines).
ANSWER. Dear colleague, we are working on this and have preliminary data.” The current draft deemphasizes the molecular evidence, so I think this is less of an issue.
Discussion: Line 2649: “(genuine mucron with conoid, .” Missing “)”

Figures (7-9): I know the other reviewer suggested that accession numbers be added to the tree, but I think the other tree looks cleaner (without numbers). I agree that these need to occur somewhere in the MS, but as a personal choice I would have put in the supplementary figures. However, given this is preference, I will yield to you as to best present these data.

Experimental design

As mentioned above, the authors have addressed my concerns in the current draft and rebuttal.

Validity of the findings

I remain supportive of these data and still recommend acceptance for publication. However, I do not think there are substantive changes to warrant me seeing additional drafts.

Additional comments

Again, I welcome these data into the literature and hope these morphological data encourage discourse and testing of these interesting hypotheses.

---

## Round 0.3 · Major Revisions

· Academic Editor

Major Revisions

The reviewer and you seem to be stuck in a circle of sorts. However, the reviewer has in this latest review provided very detailed and concrete explanations of their thoughts and reasoning for their request for revisions (see attached PDF). Either follow their comments, or if you choose not too, I would like to hear from you a detailed and scientific response as to why not. Without either a major revision or a more compelling argument from the authors, I will make a final decision upon your next submission.

Reviewer 1 ·

Basic reporting

see pdf

Experimental design

see pdf

Validity of the findings

see pdf

Additional comments

see pdf

Annotated reviews are not available for download in order to protect the identity of reviewers who chose to remain anonymous.

---

## Round 0.4 · Minor Revisions

· Academic Editor

Minor Revisions

I have heard back from reviewer 1 - and many of their comments come down to disagreements between them and your group on wording and semantics, in my opinion. While I have read and agree with much of your rebuttal, I do agree with reviewer 1 that things should be more "hedged" (or toned down) on some of your conclusions.

Specifically, these changes should help improve your work:
1. Change the title please. As stated in your rebuttal letter, you say "phylogeny" does not mean only molecular ("We would like to emphasize that when we are speaking about ‘phylogeny’, we mean the direct sense of the word: the history of a kind of organisms"), but you state "molecular phylogeny" in the title, making it look like the second use of "phylogeny" is molecular as well. In light of the reviewer's comments, this title is misleading and can be easily changed.
2. Throughout the text, please either qualify "phylogeny" when using the word (as you state in your rebuttal letter), or hedge your sentences that do talk about molecular phylogeny, as reviewer 1 requests.
Specifically, "phylogeny" on line 133 - is this the "entire" meaning or "molecular"? As well, on line 1096 you only say "phylogeny" yet say the solution is concatenated DNA - this must be "molecular phylogeny" unless I am mistaken?
3. Please go over the text one last time, and ensure that the work is clear and unambiguous in what you are trying to state. A final check should eliminate misunderstandings and ambiguous phrases.

Please contact me should you need any more advice, and I look forward to seeing a revised version.

Reviewer 1 ·

Basic reporting

see pdf

Experimental design

see pdf

Validity of the findings

see pdf

Annotated reviews are not available for download in order to protect the identity of reviewers who chose to remain anonymous.

---

## Round 0.5 · accepted · Accept

· Academic Editor

Accept

The authors have well addressed the comments from the previous round, and this manuscript is now ready to be published. I look forward to seeing this work online.